# Increasing Coverage and Precision of Textual Information in Multilingual Knowledge Graphs

**Simone Conia**[*]
Sapienza University of Rome
simone.conia@uniroma1.it

**Min Li**
Apple
min_li6@apple.com

**Daniel Lee**[*]
University of Calgary
daniel.lee1@ucalgary.ca

**Umar Farooq Minhas**
Apple
ufminhas@apple.com

**Ihab Ilyas**
Apple
iilyas@apple.com

**Yunyao Li**
Apple
yunyaoli@apple.com

## Abstract

Recent work in Natural Language Processing and Computer Vision has been using textual information – e.g., entity names and descriptions – available in knowledge graphs to ground neural models to high-quality structured data. However, when it comes to non-English languages, the quantity and quality of textual information are comparatively scarce. To address this issue, we introduce the novel task of automatic Knowledge Graph Enhancement (KGE) and perform a thorough investigation on bridging the gap in both the quantity and quality of textual information between English and non-English languages. More specifically, we: i) bring to light the problem of increasing multilingual coverage and precision of entity names and descriptions in Wikidata; ii) demonstrate that state-of-the-art methods, namely, Machine Translation (MT), Web Search (WS), and Large Language Models (LLMs), struggle with this task; iii) present M-NTA, a novel unsupervised approach that combines MT, WS, and LLMs to generate high-quality textual information; and, iv) study the impact of increasing multilingual coverage and precision of non-English textual information in Entity Linking, Knowledge Graph Completion, and Question Answering. As part of our effort towards better multilingual knowledge graphs, we also introduce WikiKGE-10, the first human-curated benchmark to evaluate KGE approaches in 10 languages across 7 language families.

## 1 Introduction

The objective of a knowledge graph is to encode our collective understanding of the world in a well-defined, structured, machine-readable representation (Hogan et al., 2021). At a high level, each node of a knowledge graph usually represents a concept (e.g., *universe*, *weather*, or *president*) or an entity (e.g., *Albert Einstein*, *Rome*, or *The Legend of Zelda*), and each edge between two nodes

is a semantic relationship that represents a fact (e.g., "*Rome* is the capital of *Italy*" or "*The Legend of Zelda* is a *video game series*"). With the wealth of information that knowledge graphs provide, they play a fundamental role in a multitude of real-world scenarios, touching many areas of Artificial Intelligence (Nickel et al., 2016), including Natural Language Processing (Schneider et al., 2022), Computer Vision (Marino et al., 2017), Information Retrieval (Reinanda et al., 2020), and recommender systems (Guo et al., 2022).

Over the years, knowledge graphs have mainly been adopted as a rich source of human-curated relational information to enhance neural-based models for tasks of varying nature (Huang et al., 2019; Bevilacqua and Navigli, 2020; Orr et al., 2021). However, ever since natural language text has proven to be an effective interface between structured knowledge and language models (Guu et al., 2020; Petroni et al., 2019; Peng et al., 2023a), the value of knowledge graphs has become twofold: besides providing relational information, knowledge graphs have also become a reliable source of high-quality textual information. Indeed, recent approaches have been increasingly reliant on textual information from knowledge graphs to surpass the state of the art (Barba et al., 2021; Chakrabarti et al., 2022; De Cao et al., 2022; Xu et al., 2023).

Unfortunately, when it comes to non-English languages, the condition of multilingual textual information in knowledge graphs is far from ideal. Indeed, popular resources present a significant gap between English and non-English textual information, hindering the capability of recent approaches to scale to multilingual settings (Peng et al., 2023b) Importantly, this gap exists in high-resource languages even if we consider basic textual properties, such as entity names and entity descriptions. The nature of the problem is dual: disparity in *coverage*, as the quantity of textual information available in

---

[*]Work done as intern at Apple.

non-English languages is more limited, and *precision*, as the quality of non-English textual information is usually lower.

In this paper, we address the aforementioned coverage and precision issues of textual information in multilingual knowledge graphs via a data-centric approach. Our contributions include the following:

- We introduce the task of automatic Knowledge Graph Enhancement (KGE) to tackle the disparity of textual information between English and non-English languages in multilingual knowledge graphs;

- We present **WikiKGE-10**, a novel human-curated benchmark for evaluating KGE systems for entity names in 10 typologically diverse languages: English, German, Spanish, French, Italian, Simplified Chinese, Japanese, Arabic, Russian, and Korean;

- We investigate how well Machine Translation (MT), Web Search (WS), and Large Language Models (LLMs) can narrow the gap between English and non-English languages.

- We propose **M-NTA**, a novel unsupervised approach, which combines MT, WS, and LLMs to mitigate the problems that arise when using each system separately;

- We demonstrate the beneficial impact of KGE in downstream tasks, including Entity Linking, Knowledge Graph Completion, and Question Answering.

We deem that achieving parity of coverage and precision of textual information across languages in knowledge graphs is fundamental to enable better and more inclusive multilingual applications. In the hope that our contributions can set a stepping stone for future research in this field, we release WikiKGE-10 at https://github.com/apple/ml-kge.

## 2 Related Work

In this section, we provide a brief overview of knowledge graphs, highlighting how textual information from knowledge graphs is now as important as their relational information, showcasing how recent work has successfully integrated textual information into downstream applications, and reviewing how recent efforts have mainly focused on completing relational information in knowledge graphs rather than textual information.

**Knowledge graphs.** Even though their exact definition remains contentious, knowledge graphs are usually defined as "*a graph of data intended to accumulate and convey knowledge of the real world, whose nodes represent entities of interest and whose edges represent potentially different relations between these entities*" (Hogan et al., 2021). Over the years, research endeavors in knowledge graphs have steadily focused their efforts primarily on using their relational information, i.e., the semantic relations between entities. Besides foundational work on knowledge graph embedding techniques, which represent the semantics of an entity by encoding its graph neighborhood (Wang et al., 2017), relational knowledge has been successfully employed in Question Answering to encode properties that generalize over unseen entities (Bao et al., 2016; Zhang et al., 2018; Huang et al., 2019), in Text Summarization to identify the most relevant entities in a text and their relations (Huang et al., 2020; Ji and Zhao, 2021), in Entity Linking to condition the prediction of an instance on knowledge subgraphs (Raiman and Raiman, 2018; Orr et al., 2021), and in Word Sense Disambiguation to produce rich meaning representations that can differentiate closely related senses (Bevilacqua and Navigli, 2020; Conia and Navigli, 2021).

**Textual information in knowledge graphs.** While knowledge graphs have been used for the versatility of their relational information, the rapid emergence of modern language models has also represented a turning point in how the research community looks at knowledge graphs. As a matter of fact, the initial wave of Transformer-based language models (Devlin et al., 2019; Radford et al., 2019) were trained purely on text, and, when researchers realized that quantity and quality of training data are two essential factors to enable better generalization capabilities (Liu et al., 2019), it became clear that the textual data available in knowledge graphs could be exploited as a direct interface between human-curated structured information and language models.

Indeed, prominent knowledge graphs – Wikidata (Vrandečić and Krötzsch, 2014), DBPedia (Lehmann et al., 2015), YAGO (Hoffart et al., 2011), and BabelNet (Navigli et al., 2021), among others – feature lexicalizations for each entity in multiple languages, e.g., names, aliases and descriptions of various length. Therefore, textual information in knowledge graph is now as important

as relational information, with recent developments taking advantage of the former to surpass the previous state of the art in an increasingly wide array of tasks, such as Word Sense Disambiguation (Barba et al., 2021), Entity Linking (Xu et al., 2023; Procopio et al., 2023), Relation Alignment (Chakrabarti et al., 2022), and Language Modeling itself (Xiong et al., 2020; Agarwal et al., 2021; Li et al., 2022a; Liu et al., 2022). Unfortunately, the wide adoption of such techniques in multilingual settings has been strongly limited by the disparity in coverage and quality of entity names and descriptions in multilingual knowledge graphs between English and non-English languages (Peng et al., 2023b).

**Knowledge graph acquisition and completion.** Finally, we would like to stress that our endeavor is orthogonal to the efforts that usually fall under the umbrella terms of "knowledge acquisition" (Ji et al., 2022) and "knowledge graph completion" in the literature (Lin et al., 2015; Shi and Weninger, 2018; Chen et al., 2020b). More specifically, the objective of these two tasks is to construct the "structure" of a knowledge graph, i.e., identifying the set of entities of interest and the (missing) relations between entities. Therefore, the multilingual extensions of these two tasks are concerned about detecting missing nodes or edges in a multilingual knowledge graph (Chen et al., 2020a; Huang et al., 2022; Chakrabarti et al., 2022), whereas we specifically focus on expanding the coverage and precision of *textual information* in multilingual knowledge graphs. Nonetheless, we argue that increasing coverage and quality of textual information in multilingual knowledge graphs has beneficial cascading effects on tasks like knowledge graph completion, as our experiments show in Section 6.

## 3 Knowledge Graph Enhancement of Textual Information

While relational information in knowledge graphs is usually language-agnostic (e.g., "AI" is a field of "Computer Science" independently of the language we consider), textual information is usually language-dependent (e.g., the lexicalizations of "AI" and "Computer Science" vary across languages). With the growing number of languages supported by knowledge graphs, it is increasingly challenging for human editors to maintain their content up-to-date in all languages: therefore, we believe it is important to invest in the development and evaluation of systems that can support humans

in updating textual information across languages.

### 3.1 Task definition

Given an entity $e$ in a knowledge graph $G$, we define Knowledge Graph Enhancement (KGE) as the task of automatically producing textual information about $e$ for each language $l \in L$, where $L$ is the set of languages of interest. More precisely, KGE encompasses two subtasks:

- Increasing **coverage** of textual information, which consists in providing textual information that is currently unavailable for $e$ in $G$;

- Increasing **precision** of textual information, which consists in identifying inaccurate or under-specified facts in the textual information already available for $e$ in $G$.

Therefore, KGE evaluates the capability of a system to provide new textual information (coverage) as well as its capability to detect errors and inaccuracies in existing textual information (precision). While textual information may refer to any entity property expressed in natural language, in the reminder of this paper, we focus on entity names and entity descriptions in Wikidata, which have become increasingly used in knowledge-infused language models and state-of-the-art systems (see Section 2).

### 3.2 Coverage of non-English information

Ideally, we would like every entity $e$ in Wikidata to be "covered" in all languages, i.e., we would like Wikidata to provide a name and a description of $e$ for each $l$ in the set $L$ of the languages supported by the knowledge graph. In practice, this is not the case in Wikidata, as we can observe in Figure 1, which provides a bird's-eye view on the availability of entity names and entity descriptions in 9 non-English languages. More precisely, we analyzed the Wikidata entities that have an associated Wikipedia page[1] with at least 100 page views in any language over the 12 months between May 2022 and April 2023. Our analysis calls attention to the issue of coverage of entity names and entity descriptions in Wikidata, which is significant even if we only consider head entities – top-10% of the most popular entities sorted by number of Wikipedia page views – and restrict the set of languages to German, Spanish, and French, which are

---

[1]The Wikidata-to-Wikipedia mapping is n-to-1 since a Wikidata entity may refer to the entire Wikipedia article or a section of an article.

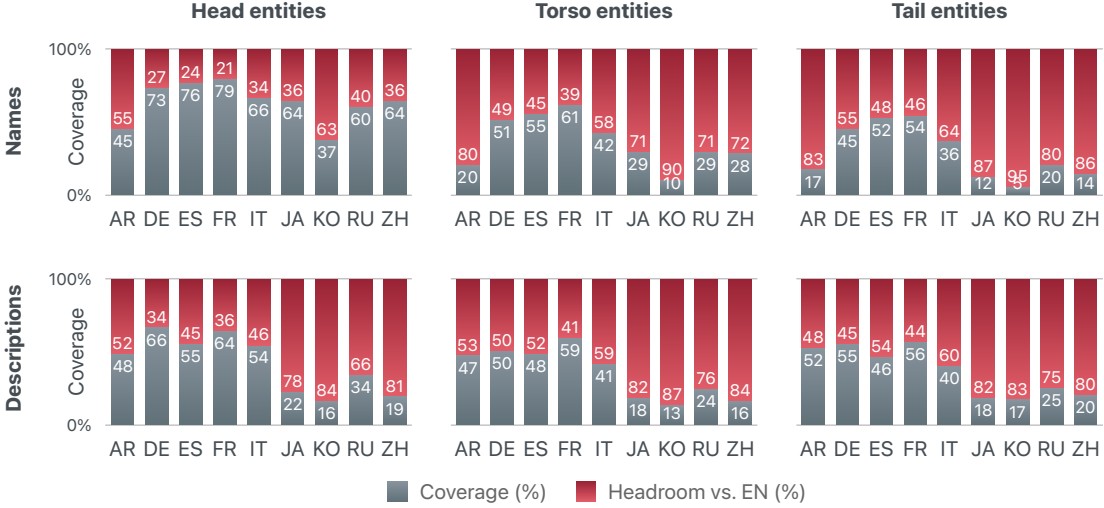

Figure 1: Coverage of non-English textual information – entity **names** and **descriptions** – compared to English in Wikidata. Even for head entities (top-10% in terms of Wikipedia page views), there is a large disparity between English and non-English coverage; the situation is unexpectedly worse on torso (top-50%) and tail entities. Best seen in color.

usually regarded as "high-resource" languages. Unsurprisingly, we can observe that the gap in coverage increases when we consider entities belonging to the torso (top-50%) and tail of the popularity distribution, as the coverage of Japanese and Chinese names for tail entities is lower than 15%.

We argue that the fact that Wikidata inherits this disparity from Wikipedia, which is edited by a disproportionate number of English-speaking contributors,[2] should not detract our attention from this issue. As a matter of fact, a growing number of approaches relies on textual information from Wikidata; therefore, we believe that the stark contrast between today's great interest for textual information in knowledge graphs and the scarce multilingual coverage revealed by our analysis motivates the development of "data-centric AI" approaches (Zha et al., 2023) for increasing multilingual coverage, rather than focusing our efforts exclusively on model-centric novelties.

### 3.3 Precision of non-English information

While non-English coverage of entity names and entity descriptions is critical, another crucial aspect is the level of precision in Wikidata. Indeed, the majority of the approaches that rely on names and descriptions often use such information as-is and overlook the possibility that it may be inaccurate. More specifically, we categorize the causes of

inaccurate information into three main classes:

- **Human mistakes**, when the imprecision was caused by a human editor. For example, entity Q1911 is incorrectly named *Oliver Giroud* in Spanish instead of *Olivier Giroud*.

- **Stale entries**, when new information is available but Wikidata has not been updated. For example, the English description of entity Q927916 has been recently updated to include the date of death but the Russian description still indicates the date of birth only.

- **Under-specific information**, when the available information is not incorrect but it is still too generic. For example, the Spanish description for Q345494 is "*músico japonés*" (Japanese musician), whereas the German one is "*japanischer Komponist, Pianist, Produzent und Schauspieler (1952–2023)*," which details his work (composer, pianist, producer, and actor) and includes his birth and death dates.

Although it is not uncommon to encounter instances of these three classes of error in Wikidata, conducting a comprehensive analysis of its entire knowledge graph is unfeasible.

### 3.4 Evaluating KGE with WikiKGE-10

To address the above-mentioned issues, we present **WikiKGE-10**, a novel resource for benchmarking data-centric-AI approaches on KGE of entity

---

[2]The primary language of Wikipedia editors is English (52%), followed by German (18%), Russian and Spanish (both at 10%) [source: UNU-Merit].

| | AR | DE | EN | ES | FR | IT | JA | KO | RU | ZH | All |
|---|---|---|---|---|---|---|---|---|---|---|---|
| Entities | 1,000 | 1,000 | 1,000 | 1,000 | 1,000 | 1,000 | 1,000 | 1,000 | 1,000 | 1,000 | 10,000 |
| Entity names in WikiKGE-10 | 4,213 | 3,498 | 2,837 | 4,320 | 3,548 | 3,156 | 2,999 | 3,874 | 3,901 | 4,088 | 36,434 |
| - Entity names in Wikidata | 2,521 | 2,336 | 2,090 | 2,732 | 2,330 | 1,840 | 2,235 | 2,136 | 2,706 | 2,569 | 23,495 |
| - Entity name errors in Wikidata | 320 | 491 | 219 | 571 | 530 | 236 | 486 | 329 | 507 | 830 | 4,663 |

Table 1: Overview of WikiKGE-10, which features 10 languages – Arabic (AR), German (DE), English (EN), Spanish (ES), French (FR), Italian (IT), Japanese (JA), Russian (RU), simplified Chinese (ZH).

names in 10 languages: English, German, Spanish, French, Italian, Chinese, Japanese, Korean, Arabic, and Russian. At a high level, WikiKGE-10 is designed to feature typologically-different linguistic families, from West Germanic to Romance, Semitic, Slavic, Koreanic, Japonic, and Sino-Tibetan, and, therefore, to enable comparison of entity names across a set $L$ of 10 diverse languages with heterogeneous, possibly non-overlapping vocabularies and scripts.

Given a language $l \in L$, we uniformly sampled 1000 entities from the top-10% of the entities in Wikidata sorted by the number of page views for their corresponding Wikipedia article in $l$. We note that the composition of the top-10% entities – and, therefore, our sample of 1000 entities – may significantly vary from language to language, as the popularity distribution changes according to what different cultures care about (Hershcovich et al., 2022). After selecting 1000 entities for each language, human graders manually checked their existing names to assess their correctness, while also adding new valid names. The annotation process, which took more than 2,500 human hours, resulted in around 36,000 manually-curated names across 10 languages; we provide more details on the creation of WikiKGE-10 and our guidelines in Appendix A. Importantly, as shown in Table 1, we find that human graders deemed 20% of the entity names in Wikidata to be incorrect and that 40% of the valid entity names could not be found in Wikipedia. In practice, since WikiKGE-10 features manually-curated entity names and indicates which names in Wikidata are incorrect or inaccurate, our benchmark can be used to evaluate the capability of a system to tackle both subtasks in KGE, i.e., increasing coverage and precision of entity names.

## 4 Methodology

In this section, we consider three broad families of approaches – MT, WS, and LLMs – and demonstrate their unsatisfactory performance on narrowing the coverage and precision gap between English and non-English languages. Therefore, we also introduce **M-NTA** (*Multi-source Naturalization, Translation, and Alignment*), a simple unsupervised ensembling technique, which overcomes the limitations of MT, WS, and LLMs by combining and ranking their predictions. Here, we direct our attention toward entity names, but we also show that the methodologies discussed in this section can be extended to other types of textual information, such as entity descriptions, in Appendix C.

### 4.1 Baseline approaches

**Machine Translation (MT).** When in need of converting information from one language to another, employing MT is a typical choice. Indeed, given a source language $l_s$ and a target language $l_t$, a straightforward approach would be to use an MT system to translate the textual information available in $l_s$ to $l_t$ to increase coverage in $l_t$. However, such an approach is limited in several respects: i) it assumes that all textual information is available in $l_s$, which, in practice, is not the case even when $l_s$ is English, i.e., MT cannot be applied if the information to translate is not available in the source language in the first place; ii) it assumes that MT systems are precise, which, again, is not the case: for example, entity names can be complex and ambiguous to translate without additional context (e.g., "Apple" could refer to the fruit or the tech company); and, iii) while MT can be employed to increase coverage, it is not clear how to apply MT to identify inaccurate entity names to increase precision of existing textual information.

**Web Search (WS).** A common workflow for looking up textual information in a target language $l_t$ is to query Web search engines with queries in a source language $l_s$, such as "*[entity-name]* in *[l_t]*", and extract the answer from the search results, possibly limiting the search space to Web pages entirely in $l_t$ or originating from countries in which $l_t$ is the primary/official language. While

WS can provide more varied results that are not 1-to-1 translations of the source entity name, we argue that WS suffers from the same fundamental limitations as MT: i) if $l_s$ is not complete, then we cannot formulate every search query; ii) WS is prone to biases, especially for ambiguous instances (e.g., googling "plane" shows many results about airplanes, a few results about geometric planes, and none about plane trees); and, iii) using WS to identify and correct imprecise textual information in a knowledge graph is not obvious.

**Large Language Models (LLMs).** Recent LLMs have been shown to be few-shot learners, thanks to what is now known as in-context learning, or the capability of capturing latent relationships between a few input examples to provide an answer for a new task (Brown et al., 2020). With the advent of multilingual LLMs, such as BLOOM (Scao et al., 2023), mT5 (Xue et al., 2021), and their instruction-fine-tuned variants (Muennighoff et al., 2022), we can prompt such models for translation, e.g., "How do you say *[entity-name]* in *[$l_t$]*?", possibly providing a few examples in input to condition the generation of the output. While prompting language models is versatile, relying on LLMs also exposes us to their weaknesses, e.g., hallucinations (Ji et al., 2023) and data biases (Navigli et al., 2023).

### 4.2 M-NTA: Multi-source Naturalization, Translation, and Alignment

To address the issues above, we introduce M-NTA, a simple unsupervised technique that combines MT, WS, and LLMs. The intuition behind M-NTA is that obtaining a fact from multiple source systems may offer complementary pieces of information which provide varying *views* on our world knowledge; we hypothesize that, if distinct views support the same fact, there is a greater chance for the fact to be closer to the ground truth.

**Source systems in M-NTA.** The first question, therefore, is how to produce the above-mentioned views on our world knowledge. Given a source language $l_s$ and an entity $e$ whose name in $l_s$ is $e_s^n$, M-NTA takes a three-steps approach to generate $e_t^n$ in a target language $l_t$:

1. **Naturalization:** as mentioned above, entity names are not suitable for direct translation since they might not provide sufficient context (Li et al., 2022b). To overcome this issue, M-NTA retrieves the textual description $e_s^d$ of

$e$ in $l_s$ from Wikidata and uses it to produce a natural language representation $r_s(e_s^n, e_s^d)$ of $e$ in $l_s$. This allows M-NTA to rely on different representations for polysemous words, e.g., "*Apple* is an *American technology company*" and "*Apple* is a *fruit of the apple tree*."

2. **Translation:** next, M-NTA "translates" the representation $r_s(e_s^n, e_s^d)$ from $l_s$ to $l_t$ using a system $f(\cdot)$ to obtain a natural language output $r_t(e_t^n, e_t^d)$ in the target language.

3. **Alignment:** finally, M-NTA aligns the output $r_t(e_t^n, e_t^d)$ with the input $r_s(e_s^n, e_s^d)$ to extract the entity name $e_t^n$.

Most crucially, M-NTA is transparent to the definition of a source system $f(\cdot)$. This allows M-NTA to take advantage of any source system $f(\cdot)$ that is able to produce $e_t^n$. More specifically, M-NTA can use a set of source systems $F = \{f_1, f_2, \ldots, f_n\}$ in which $f_i$ can be an MT, WS or LLM-based system. Not only that, we can leverage the same MT system multiple times by setting the source language $l_s$ to different languages, allowing M-NTA to draw knowledge from all the languages of interest to produce better results in $l_t$.

**Ranking answers in M-NTA.** The second question is how to validate each view by using the other views. In practice, we first consider each view as an answer $y = f(\cdot)$ provided by a source system $f(\cdot)$ in the set of source systems $F$. Then, we assign an agreement score $\sigma(y)$ to each answer:

$$\sigma(y) = \sum \phi(y, y') \quad \forall y' = f'(\cdot), f' \in F \setminus \{f\}$$

where $\phi(y, y') \to \{0, 1\}$ is a function that indicates if $y$ is supported by $y'$, e.g., in the case of entity names $\phi(\cdot, \cdot)$ can be implemented as exact string match. In other words, the agreement score $\sigma(y)$ is higher when an answer $y$ from a source system $f$ is supported by an answer $y'$ from another source system $f'$; if $y$ is valid according to multiple source systems, then there is a lower chance for $y$ to be incorrect. On the contrary, if $y$ is not supported by other answers, its agreement score is lower and, therefore, there is a higher chance for $y$ to be incorrect. Finally, we obtain the final set of answers $Y$ by selecting all the answers $y$ whose score $\sigma(y)$ is greater than or equal to a threshold $\lambda$:

$$Y = \{y : \sigma(y) \geq \lambda\}$$

| | | AR | | DE | | EN | | ES | | FR | | IT | | JA | | KO | | RU | | ZH | | Avg | |
|---|---|---|---|---|---|---|---|---|---|---|---|---|---|---|---|---|---|---|---|---|---|---|---|
| | | C | P | C | P | C | P | C | P | C | P | C | P | C | P | C | P | C | P | C | P | C | P |
| *MT from* | DE → | 28.1 | 42.4 | – | – | 37.8 | 60.1 | 47.1 | 60.9 | 48.3 | 59.9 | 51.6 | 60.3 | 24.1 | 52.1 | 30.8 | 46.3 | 36.2 | 54.6 | 28.3 | 54.8 | 36.9 | 54.6 |
| | EN → | 30.2 | 45.1 | 52.1 | 67.1 | – | – | 50.9 | 63.1 | 50.2 | 62.8 | 54.1 | 65.2 | 29.9 | 55.3 | 32.8 | 49.2 | 38.1 | 57.3 | 30.6 | 57.1 | 41.0 | 58.0 |
| | ES → | 27.3 | 43.1 | 48.0 | 63.1 | 37.1 | 58.5 | – | – | 48.9 | 60.8 | 52.9 | 64.0 | 27.7 | 54.1 | 32.0 | 47.4 | 36.2 | 55.2 | 27.1 | 53.2 | 37.5 | 55.5 |
| | FR → | 27.0 | 43.6 | 47.4 | 63.5 | 37.6 | 58.3 | 48.3 | 58.9 | – | – | 52.9 | 64.0 | 27.4 | 54.5 | 32.3 | 47.8 | 35.9 | 55.1 | 27.4 | 53.6 | 37.4 | 55.5 |
| | IT → | 26.8 | 43.6 | 48.2 | 62.9 | 36.4 | 58.7 | 46.8 | 57.8 | 49.2 | 61.3 | – | – | 28.0 | 54.6 | 31.6 | 48.2 | 35.9 | 55.4 | 26.3 | 53.5 | 36.6 | 55.1 |
| | JA → | 23.3 | 37.1 | 43.0 | 57.1 | 31.1 | 52.5 | 43.3 | 52.1 | 44.9 | 56.8 | 48.9 | 60.0 | – | – | 28.0 | 43.4 | 32.2 | 51.2 | 23.1 | 49.2 | 35.3 | 51.0 |
| | ZH → | 22.7 | 36.2 | 42.0 | 55.7 | 30.2 | 49.2 | 39.0 | 48.1 | 40.3 | 51.8 | 44.9 | 58.0 | 19.3 | 44.1 | 26.0 | 39.4 | 29.4 | 48.5 | – | – | 32.6 | 47.9 |
| *WS* | Google$_{Search}$ | 14.6 | 28.0 | 36.4 | 54.1 | – | – | 39.3 | 52.0 | 39.0 | 57.6 | 43.6 | 53.5 | 16.1 | 44.3 | 23.6 | 38.5 | 29.1 | 47.2 | 18.5 | 36.2 | 28.9 | 45.7 |
| *LLMs* | mT0$_{large}$ | 15.2 | 29.0 | 40.1 | 53.2 | – | – | 40.3 | 53.1 | 39.4 | 57.2 | 44.2 | 54.1 | 16.5 | 44.1 | 22.4 | 39.2 | 28.3 | 47.4 | 18.0 | 37.0 | 29.4 | 46.0 |
| | mT0$_{xl}$ | 15.8 | 31.1 | 42.1 | 54.4 | – | – | 41.5 | 54.2 | 39.9 | 58.0 | 44.5 | 54.9 | 16.9 | 46.1 | 23.2 | 39.5 | 30.1 | 48.4 | 19.2 | 37.8 | 30.4 | 47.2 |
| | mT0$_{xxl}$ | 17.1 | 33.4 | 43.8 | 56.1 | – | – | 41.9 | 55.0 | 40.7 | 59.1 | 45.0 | 55.1 | 18.0 | 46.9 | 22.4 | 39.9 | 31.0 | 48.7 | 19.3 | 40.1 | 31.0 | 48.3 |
| | GPT-3 | 18.2 | 34.1 | 47.4 | 64.9 | – | – | 45.3 | 60.2 | 45.4 | 62.2 | 49.4 | 62.2 | 21.4 | 49.1 | 26.0 | 42.7 | 32.3 | 53.5 | 22.1 | 50.8 | 34.2 | 53.3 |
| | GPT-3.5 | 27.4 | 42.1 | 50.5 | 66.2 | – | – | 50.6 | 63.2 | 50.5 | 63.3 | 53.7 | 64.9 | 28.9 | 54.4 | 31.9 | 47.3 | 36.8 | 56.3 | 29.2 | 55.7 | 39.9 | 57.0 |
| | GPT-4 | 29.9 | 44.0 | 51.3 | 66.1 | – | – | 50.7 | 63.0 | 51.4 | 63.6 | 54.7 | 65.6 | 33.7 | 56.3 | 34.6 | 48.9 | 40.2 | 58.5 | 31.3 | 56.5 | 42.0 | 58.1 |
| *M-NTA* | M-NTA$_{GPT-3}$ | 41.3 | 73.6 | 57.5 | 77.3 | 41.3 | 64.8 | 55.4 | 74.7 | 57.1 | 69.9 | 61.3 | 75.1 | 34.0 | 65.8 | 50.0 | 76.6 | 44.1 | 66.5 | 34.7 | 70.0 | 53.0 | 79.4 |
| | M-NTA$_{GPT-3.5}$ | 42.7 | **74.4** | 57.5 | 77.6 | **41.3** | **64.8** | 55.6 | 75.0 | 57.3 | 70.0 | **61.7** | 75.2 | **35.2** | 67.0 | 50.6 | 76.7 | 44.8 | 66.9 | 36.1 | 71.4 | 53.6 | 79.9 |
| | M-NTA$_{GPT-4}$ | **43.2** | **74.4** | 57.1 | 77.5 | **41.3** | **64.8** | **55.8** | 75.0 | **57.4** | 70.3 | **61.7** | **75.5** | **35.2** | **67.9** | **51.2** | **76.8** | **45.3** | **67.1** | **36.8** | **72.0** | **53.9** | **80.1** |

Table 2: F1 scores on entity names coverage (C) and precision (P) in WikiKGE-10 for MT with NLLB-200, WS with Google Search, LLM prompting with mT0 and GPT, and M-NTA. The symbol "–" is used to indicate that source and target languages are the same. Best results in **bold**.

where $\lambda$ is a hyperparameter that can be tuned to balance precision and recall of the system, with our experiments indicating that $\lambda = 2$ is the most balanced choice for coverage, as discussed in Appendix C.

Differently from MT, WS, and LLMs, since each answer in $Y$ is scored and ranked by M-NTA, the application of M-NTA to KGE is straightforward. To increase coverage, we can consider $Y$ as the result, as $\lambda > 1$ allows M-NTA to remove unlikely answers; to increase precision, we can consider every value $\hat{y}$ in the KG that is not in $Y$ as an incorrect value.

## 5 Experiments on KGE

In this section, we evaluate our strong baselines and M-NTA on the task of KGE for entity names and discuss the results obtained on WikiKGE-10.

### 5.1 Experimental setup

Recently, there has been a surge of interest for multilingual MT systems, i.e., systems that use a unified model for multiple language pairs. Therefore, for the implementation of the MT baseline, we use NLLB-200 (Costa-jussà et al., 2022), a state-of-the-art multilingual MT system that supports over 200 languages. For WS, we use Google Web Search, as it is often regarded as one of the best WS engines. For LLM prompting, we consider two popular models: i) mT0 (Muennighoff et al., 2022), an openly available instruction-finetuned multilingual LLM based on mT5, and ii) GPT,[3] one of the most pop-

ular albeit closed LLMs, which has been proven to show strong multilingual capabilities. Finally, we evaluate M-NTA when scoring and ensembling the outputs from NLLB-200,[4] Google Web Search, and GPT-3/3.5/4.

For each baseline, the input data is the set of entity names that currently exist in Wikidata in a source language $l_s$, i.e., the entity names in $l_s$ are "translated" into the target language $l_t$ using MT, WS, LLM prompting or M-NTA. We note that, if Wikidata does not include at least one name for an entity $e$ in $l_s$, then none of the systems mentioned above is able to produce a name in $l_t$. M-NTA is able to mitigate this issue by drawing information from multiple source languages at the same time.

Given a set of human-curated correct names $\bar{Y}$ from WikiKGE-10 and a set of predicted names $Y$ generated by a system, we compute coverage between $\bar{Y}$ and $Y$ as following:

$$\text{PPV}_C = \sum_{y \in Y} \frac{\mathbb{1}_{\bar{Y}}(y)}{|Y|}$$

$$\text{TPR}_C = \sum_{\bar{y} \in \bar{Y}} \frac{\mathbb{1}_Y(\bar{y})}{|\bar{Y}|}$$

$$\text{Coverage} = 2 \frac{\text{PPV}_C \cdot \text{TPR}_C}{\text{PPV}_C + \text{TPR}_C}$$

where $\text{PPV}_C$ is the positive predictive value, $\text{TPR}_C$ is the true positive rate, and $\mathbb{1}_X(x)$ is the indicator function, which returns 1 if $x \in X$ else 0. We compute precision in a similar way, using the

---

[3]Experiments with GPT-3 and GPT-3.5 were carried out between March and May 2023. Additional experiments with

GPT-4 were carried out in September 2023.

[4]For each target language $l_t$, we M-NTA uses the translations from every source language $l_s \neq l_t$.

set of human-curated invalid names $\neg \bar{Y}$ and the set of names $\neg Y$ predicted to be incorrect by a system. Note that, to enable a direct and fair comparison, we allow every system to rely on additional contextual information in the form of entity descriptions from Wikidata; we provide more details about the experimental setting in Appendix C.

## 5.2 Results and discussion

The results on WikiKGE-10 reported in Table 2 highlight two key findings: i) our proposed solution, M-NTA, offers superior performance compared to state-of-the-art techniques in MT, WS, and LLMs on both coverage and precision of entity names; and, ii) the results on WikiKGE-10 indicate that KGE is a very challenging task and that more extensive investigations are needed to design better KGE systems. In the following, we report the main takeaways from our experiments.

**Different languages hold different knowledge.** Our experimental results show that generating entity names in non-English languages by translating English-only textual information does not provide the best results, as shown in Table 2. This is true not only for the MT system we use in our experiments but also for WS and LLMs, for which we use English-only queries and prompts, respectively. In particular, it is interesting to notice that completely different systems, namely, MT and GPT-3.5, produce similar results on average: 41.0% vs. 39.9% in coverage and 58.0% vs. 57.0% in precision. Therefore, we hypothesize that the significant gain in performance by M-NTA – +12% in coverage and +22% in precision over GPT-3.5 – is mainly attributable to its effectiveness in combining information across different languages. Indeed, it is interesting to notice that using GPT-4 instead of GPT-3.5 as one of the sources of M-NTA only provides marginal improvements to the overall results in both coverage and precision.

**WS may not be suitable for KGE.** The results from our experiments show that WS is the least effective approach to generate entity names. Although we are not disclosed on the inner workings of proprietary search engines, we can qualitatively observe that the results returned from Web searches often include answers for entities that are semantically similar to the one mentioned in the input query. For example, searching *Niki Lauda* (former F1 driver) in Italian also returns results about *Rush* (biographical film on Lauda). Relying on semantic similarity is often a robust strategy for information retrieval, but, in this case, it introduces significant noise, which is undesirable in a knowledge graph.

**Prompting LLMs requires caution.** Our experiments also indicate that prompting LLMs is a better option than WS in terms of performance, especially when using GPT. However, we shall keep in mind not to take benchmark results at face value (Maru et al., 2022): analyzing the answers shows one issue that does not surface in our numerical results is that some errors in the predictions provided by LLMs can be significantly worse – and, therefore, potentially more problematic – than those made by MT and WS systems. We observe that, especially for uncommon entities and smaller models, LLMs may produce answers that are completely unrelated to the correct answer, including copying part of the prompt or its examples, providing entity names for entirely different entities (e.g., *Silvio Berlusconi* (Italian politician) for *San Cesario sul Panaro* (Italian comune)), hallucinating facts (e.g., adding that *The Mandalorian (2nd season)* is from *Star Wars: La venganza de los Sith* in Spanish), and also generating nonsense outputs. It follows that, although LLMs are generally better than WS, the risk of using them is higher in case of error, as purely numerical metrics, such as coverage and precision, may hide that some errors are worse than others, i.e., potentially more harmful in downstream applications.

## 6 Enhancing Textual Information in KGs: Impact on Downstream Tasks

In this section, we demonstrate the beneficial impact of KGE on downstream tasks and its effectiveness in improving the performance of state-of-the-art techniques in multilingual Entity Linking and Knowledge Graph Completion; we also show that KGE is beneficial for multilingual Question Answering in Appendix E.

**Multilingual Entity Linking (MEL).** A direct application of increasing the quantity and quality of textual information in a knowledge graph is MEL, the task of linking a textual mention to an entity in a multilingual knowledge base (Botha et al., 2020). We evaluate the impact of our work on mGENRE (De Cao et al., 2022), a state-of-the-art MEL system that fine-tunes mBART (Lewis et al., 2020) to autoregressively generate a Wikidata entity name for a mention in context. As noted

| MEL | mGENRE | mGENRE + M-NTA | Δ |
|---|---|---|---|
| FR | 73.4 | **74.1**⋆ | +0.7 |
| IT | 56.8 | **58.2**⋆ | +1.4 |
| RU | 65.8 | **66.2**⋆ | +0.4 |
| ZH | 52.8 | **55.0**⋆ | +2.2 |
| Avg | 62.2 | **63.3**⋆ | +1.2 |

Table 3: Comparison between mGENRE and mGENRE + M-NTA in terms of F1 score in multilingual Entity Linking on Wikinews-7. Best results in **bold**. ⋆: statistically different with $p < 0.05$.

| MKGC | A-KGC | A-KGC + M-NTA | Δ |
|---|---|---|---|
| EN | 47.4 | **47.5** | +0.1 |
| ES | 64.6 | **66.3**⋆ | +1.7 |
| FR | 64.4 | **66.0**⋆ | +1.6 |
| JA | 62.8 | **64.2**⋆ | +1.4 |
| Avg | 59.8 | **61.1**⋆ | +1.3 |

Table 4: Comparison between our re-implementation of Align-KGC and Align-KGC + M-NTA in terms of Mean Reciprocal Rank (MRR) in multilingual Knowledge Graph Completion on DBP-5L. Best results in **bold**. ⋆: statistically different with $p < 0.05$.

by De Cao et al. (2022), mGENRE generates entity names by also copying relevant portions of the input mention; however, copying is not possible when the mention of the entity is in a language for which Wikidata does not feature any names. By increasing the coverage and precision of textual information in Wikidata, M-NTA provides mGENRE with a broader coverage of entity names in non-English languages, aiding mGENRE's capability to rely on copying mechanisms. Indeed, as we can see in Table 3, augmenting mGENRE with M-NTA brings an improvement of 1.2 points in F1 score on average in Wikinews-7, setting a new state-of-the-art on this benchmark.

**Multilingual Knowledge Graph Completion (MKGC).** Another direct application of KGE is MKGC, the task of predicting missing links between two entities in a multilingual knowledge base (Chen et al., 2020a). Similarly to MEL, we evaluate the downstream impact of our work on a re-implementation of Align-KGC (SoftAsym), a state-of-the-art MKGC system originally proposed by Chakrabarti et al. (2022), which we rebuilt to use our entity names and descriptions to create mBERT-based entity embeddings. As shown in Table 4, using M-NTA to provide more and better entity names and descriptions allows the MKGC system to obtain a consistent improvement across non-English languages on DBP-5L (Chen et al., 2020a), i.e., +1.5 points in terms of Mean Reciprocal Rank (MRR), excluding English. We hypothesize that the larger part of this improvement comes from the fact that the entity descriptions generated by M-NTA are more informative, as suggested by the examples shown in Appendix C.7 (see Table 7). On one hand, this improvement demonstrates the flexibility of M-NTA, as DBP-5L is based on a different knowledge graph, i.e., DBPedia. On the

other hand, it empirically validates our assumption that increasing coverage and precision of textual information in multilingual knowledge graphs is an effective data-centric way to unlock latent performance in current systems.

## 7 Conclusion and Future Work

In this paper, we introduced the novel task of automatic Knowledge Graph Enhancement, with the objective of fostering the development and evaluation of data-centric approaches for narrowing the gap in coverage and precision of textual information between English and non-English languages. Thanks to WikiKGE-10, our novel manually-curated benchmark for evaluating KGE of entity names in 10 languages, we brought to light the unsatisfactory capabilities of machine translation, web search, and large language models to bridge this multilingual gap. To this end, we introduced M-NTA, a novel approach to combine the complementary knowledge produced by the above techniques to obtain higher-quality textual information for non-English languages. Not only did M-NTA achieve promising results on WikiKGE-10 but our experiments also demonstrated its beneficial effect across several state-of-the-art systems for downstream applications, namely, multilingual entity linking, multilingual knowledge graph completion, and multilingual question answering.

We hope that our novel benchmark and method can represent a milestone for KGE. However, our work demonstrates that, if we aspire to achieve quantity and quality parity across languages, we still need more extensive investigations on how to effectively increase coverage and precision of textual information in multilingual knowledge graphs.

## Limitations

**Textual information in knowledge graphs.** In this paper, we focus on two specific types of textual information, namely, entity names and entity descriptions. Although our discussion on coverage and precision of textual information (or lack thereof) can be extended to other types of textual information, e.g., longer descriptions like Wikipedia abstracts or coreferential information like the anchor text of the hyperlinks in a Wikipedia article, our analysis in Sections 3.2 ("Coverage of non-English information") and 3.3 ("Precision of non-English information") highlights that the gap between English and non-English names and descriptions is very large even for popular entities, ranging from 20% to 60% for entity names and from 30% to 80% for entity descriptions. Furthermore, entity names and entity descriptions are the most widely used types of textual information from knowledge graphs in downstream tasks (see Section 2), and, therefore, we decided to focus our discussion on these two types, which potentially have a more direct impact on downstream applications, as also shown in Section 6. We hypothesize that most of our observations generalize to other types of textual information in knowledge graphs; however, we leave deeper investigations and the creation of benchmarks for other types of textual information in knowledge graphs to future work.

**Different knowledge graphs.** Our attention is mainly directed at Wikidata, as it is one of the most popular multilingual knowledge graphs used by the research community in Natural Language Processing as well as Information Retrieval and Computer Vision. Therefore, a possible limitation of our work is its generalizability to other knowledge graphs. We hypothesize that our work is generalizable to other knowledge graphs, such as DBPedia, BabelNet, and Open Multilingual WordNet, among others, since entity names (or aliases) and entity descriptions (or definitions) are often available in many of them. Our hunch is partially demonstrated by our empirical experiments on Multilingual Knowledge Graph Completion (see Section 6), as we evaluate the impact of M-NTA on DPB-5L, which is constructed from DBPedia. However, we hope that our work will raise awareness on the issues of multilingual coverage and precision of textual information on as many knowledge graphs as possible, and inspire future work to investigate the extent of the problem not only on general knowledge graphs but also on domain-specific ones.

**WikiKGE-10.** Although WikiKGE-10 covers a wide range of entities – a total of 36,434 manually-curated entity names – it still focuses only on entities belonging to the head of the popularity distribution of Wikipedia. Our attention is directed to popular entities as we observed a large gap of coverage between English and non-English languages even for entities that are in the top-10%: our benchmark shows that current state-of-the-art techniques, namely, MT, WS, and LLMs, still struggle to provide correct entity names for popular entities. We hypothesize that such techniques will also struggle on less popular entities, i.e., entities belonging to the torso and tail of the popularity distribution. However, we cannot assume that the performance and – more importantly – the ranking between MT, WS, and LLMs is the same on torso and tail entities, e.g., WS may be more robust than LLMs in generating names for tail entities. Future work may take advantage of the methodology presented in this paper to create benchmarks for more challenging settings. Last but not least, we stress the fact that the popularity of an entity is variable over time; therefore, entities that are now in the top-10% may not be as popular in the next year, or vice-versa, previously unknown entities may become extremely popular in the short-term future.

**M-NTA.** In Section 5.2, we demonstrate that M-NTA is able to combine information from MT, WS, and LLMs, successfully outperforming the three approaches in increasing coverage and precision of entity names across the 10 languages of WikiKGE-10. However, one of its main limitations comes from the fact that M-NTA requires the output from MT, WS, and LLMs, therefore, its inference time and computational cost is equal to the sum of its individual components if run sequentially. Since we want a knowledge graph to contain the best textual information possible, we believe that the increase in performance – +12% in terms of average F1 score on coverage increase compared to the second best system; +22% on increasing precision – justifies the additional time and compute required to run M-NTA. However, we look forward to novel methods that will be able to obtain the same or even better results while drastically decreasing the computational requirements.

## Acknowledgements

This work would not have been possible without the invaluable feedback by and conversations with Behrang Mohit, Saloni Potdar, Farima Fatahi Bayat, Ronak Pradeep, and Revanth Gangi Reddy.

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

## A    Creating WikiKGE-10

In this section, we provide more details on the creation process of WikiKGE-10, our novel human-curated dataset for evaluating automatic approaches on KGE of Wikidata entity names.

### A.1    Choice of languages

As mentioned in Section 3.4, one of the main design decision for our benchmark is the selection of 10 languages from a set of diverse typologically-different linguistic families:

- West Germanic: English, German;

- Romance: Spanish, French, Italian;

- Semitic: Arabic;

- Sino-Tibetan: Chinese (simplified);

- Slavic: Russian;

- Koreanic: Korean;

- Japonic: Japanese.

This design choice makes WikiKGE-10 challenging, as the set of symbols used in each language may or may not vary significantly: for example, a person name may be the same in English and French, but it is highly unlikely that a person name is written in the same way in English and Chinese, which requires at least transliteration. Moreover, the transliteration process between English and Chinese (and also other languages, such as Japanese) is not always deterministic, making it difficult to rely on rule-based approaches to translate a name between these two distant languages. We focused on languages that can be considered high/medium-resource as our quantitative analysis in Section 3.2 shows that coverage of textual information is still far from ideal even for the most popular entities (top-10%) of those high/medium-resource languages. We leave the expansion of our benchmark to lower-resource languages to future work.

### A.2    Human annotation process

The objective of the annotation process was to suggest and rate entity names in a target language.

First, given an entity, the human annotators were asked to familiarize themselves with its information: the user interface for the task provided the

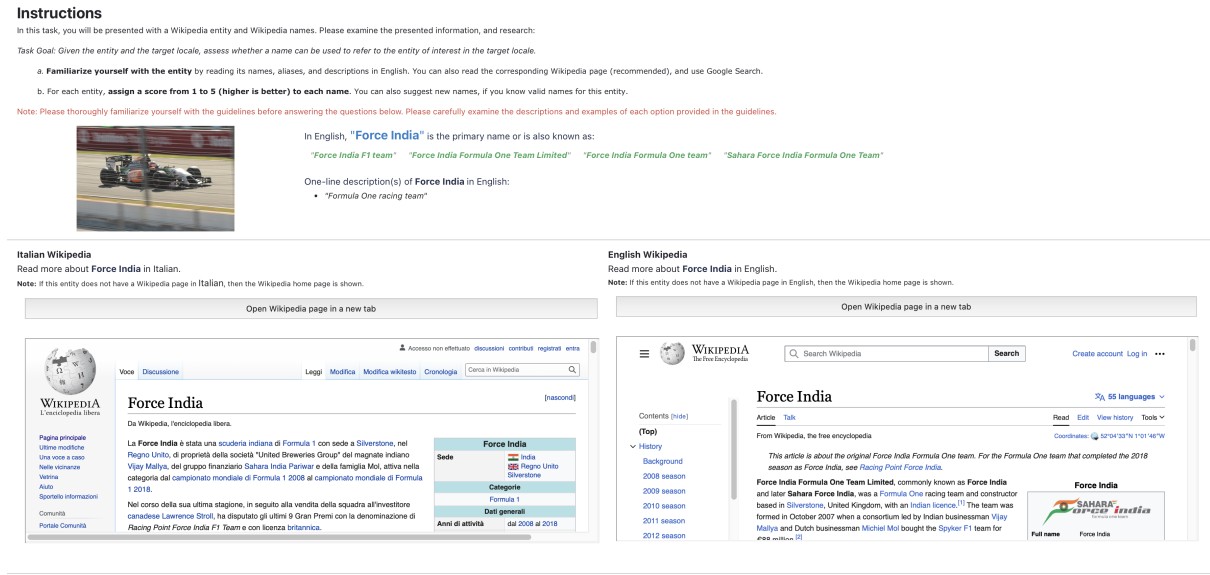

Figure 2: UI used for the annotation task: the annotators coudl familiarize themselves with the task with an outline of the task instructions (detailed guidelines could be read in a separate page) and the information about the entity, including its names in English and its Wikipedia pages in English and the target language (Italian in this case).

entity names and a short description of the given entity in English retrieved from Wikidata, as well as a built-in panel that directly displayed side-by-side the Wikipedia articles of the corresponding entity both in English and in the target language, if available. This allowed human annotators to familiarize themselves with the entity and catch commonalities and differences between English and non-English information at a glance without leaving the annotation tool.

After learning about the entity, the annotators were tasked with rating entity names that are valid for the given entity with respect to the target language, i.e., if an entity name is valid only in languages that are different from the target language of interest, the annotators were explicitly asked to categorize such names as invalid. More specifically, for each name, an annotator could choose one of the following options:

- **1 - Incorrect.** The name should not be used to refer to the entity in the target language. For example, "pomodori marci" – the literal translation of "tomatoes that are rotten (fruit)" in Italian – should never be used to refer to Rotten Tomatoes (the media review site). In addition, the name should always be valid in the target locale; therefore, a name in another language that is not recognized in the target locale should be considered incorrect.

- **2 - Spelling issues.** The name contains minor issues, for example, spelling errors or missing digits. For example, "Michael Jacson" (notice the missing "k") should not be used to refer to "Michael Jackson".

- **3 - Generic, rare or incomplete.** The name can be used to refer to this entity but it is very generic, rare or incomplete. For example, "Barack" can be used to refer to "Barack Obama" or "game" can be used to refer to "video game." Note that nicknames or stage names like "Air Jordan" for Michael Jordan (basketball player) or "Money" for Floyd Mayweather (boxer) do not fall into this category; they should be categorized as "good fit" (see below).

- **4 - Good fit.** The name is a good way to refer to this entity (for example, one of its common names, a nickname, or an acronym). For example, "Harvard" can be used to refer to "Harvard University", "WB" can be used to refer to "Warner Bros.", "Schumi" is a valid nickname for "Michael Schumacher".

- **5 - Perfect fit.** The name is the most appropriate name for this entity (usually, its most common name). For example, "Harvard University" (instead of just "Harvard"), "Barack

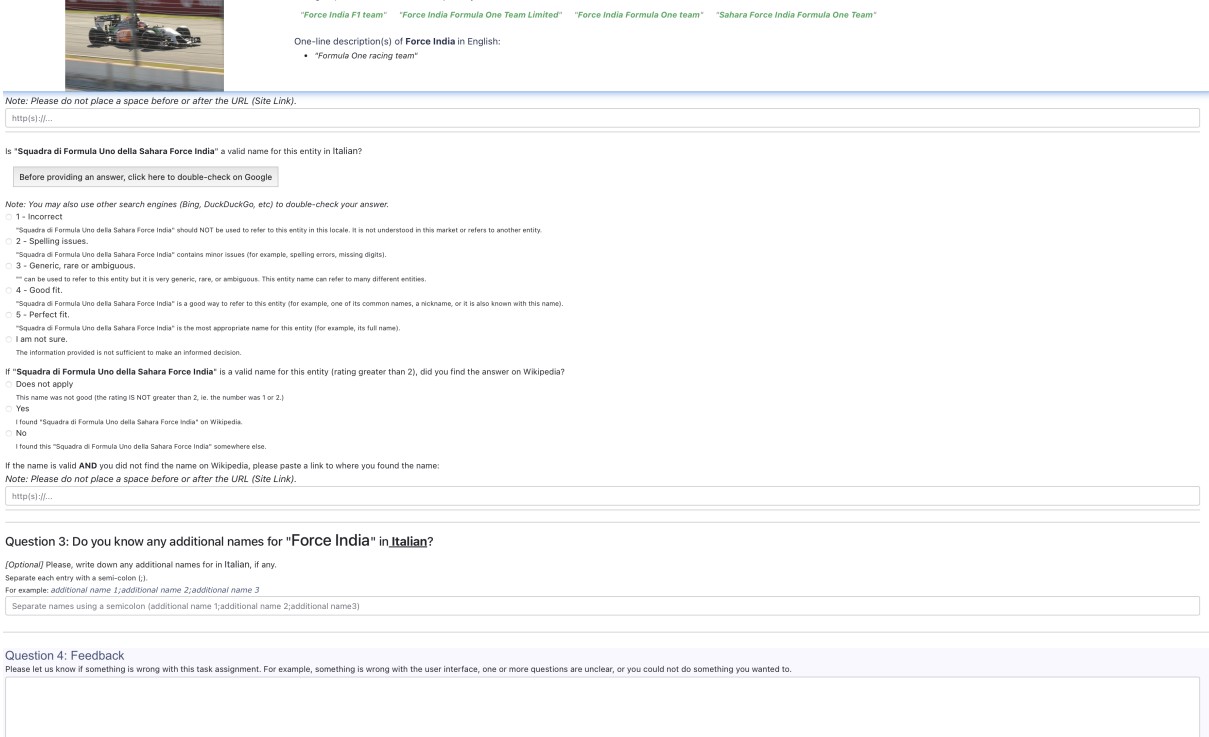

Figure 3: UI used for the annotation task: the annotator had to rate names from 1 to 5. Before providing a rating, they can easily double-check the name in consideration on a Web search engine. As shown in this figure, the annotators could also suggest new names, which will be inserted in the pool of entity names to grade.

Obama" (instead of "Barack Hussein Obama II"). In other words, it is the most common or popular entity name to reference the intended entity.

Annotators were given the choice to opt out from rating an entity name in case they deemed they did not have enough context (e.g., information from the Wikipedia pages of the entities in English and in the target language) or they did not feel knowledgeable enough about the topic.

Before confirming their selection, each annotator had to double-check their choice by searching exact matches of the name under consideration using a Web search engine; a UI component allowed the annotators to directly look up for exact matches in the target language without manually typing a query, making the search easier and speeding up the annotation process. Forcing the annotators to take this extra step allowed them to verify that a named they deemed invalid was indeed invalid, i.e., no or few results from the search engine, or not associated to the entity of interest. An example of an annotation task is shown in Figures 2 and 3.

We note that annotators could also suggest new names in the target language for each entity if they knew about other possible valid names. Each suggested name was inserted in the pool of entity names to validate, and, therefore, graded by 3 annotators. On the contrary, annotators could not suggest invalid entity names for an entity, as our objective was to focus on the errors that are already in Wikidata, but could provide feedback in case they noticed that something was wrong in the task.

### A.3 Quality assurance and inter-annotator agreement

To guarantee a high-quality output, before participating to the annotation process, each human annotator had to pass an entrance test, which consisted in studying a set of guidelines – which introduced the annotator to the concepts of entities and knowledge graphs, described the task and the UI elements, and provided a few examples with illustrations – and in rating 50 entity names correctly. Annotators that could not pass the entrance test could not participate to the actual annotation process (we did not use the 50 entity names in the entrance test in the final dataset).

For each target language, we only hired annotators that could certify their proficiency in English

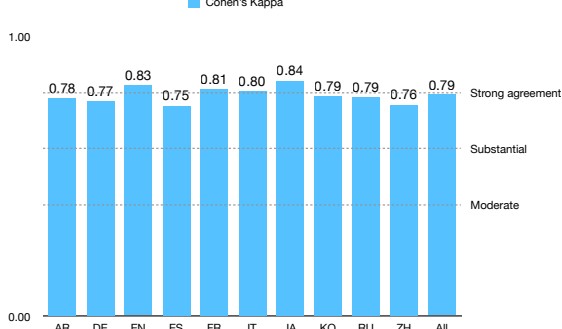

Figure 4: Pairwise inter-annotator agreement measured with Cohen's Kappa shows strong agreement at the end of the annotation process.

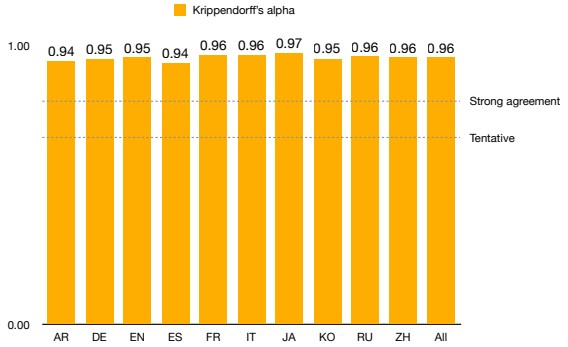

Figure 5: Inter-annotator agreement measured with Krippendorff's alpha, which takes into account the cardinality of the ratings (1-5), shows strong agreement among annotators.

and the target language. Annotators were compensated according to the standard hourly wages of their geographic location. On average, each annotator spent about 1 minute for rating an entity name and about 5 minutes on each entity. Since each entity name was rated by 3 annotators, we can estimate that the total human time required by the annotation process is 3 annotators $\times$ 10,000 entities $\times$ 5 minutes / 60 minutes = 2,500 hours.

At the end of the annotation process, we measured the inter-annotator agreement in two ways. First, we computed pairwise inter-annotator agreement using Cohen's kappa. As shown in Figure 4, we can observe an average agreement of 0.79, where a score of 0.60 is usually considered to represent substantial agreement and 0.80 is usually regarded as strong agreement. We also stress that Cohen's kappa does not take into account the cardinality of the rating values, i.e., for Cohen's kappa there is no difference between a 1-vs-5 and a 4-vs-5 disagreement. Therefore, we also measured the overall inter-annotator agreement using Krip-

pendorff's alpha, which shows strong agreement with an average of 0.96 across all languages, as we can see in Figure 5.[5] Overall, the strong inter-annotator agreement scores validate the results of the annotation process.

## B  Related Work: Addendum

While WikiKGE-10 is the first benchmark designed to aid development and evaluation of systems for increasing coverage and precision of entity names in multilingual knowledge graphs, there has been previous work that tried to address this issue in other ways. Among them, we acknowledge the existence of JRC-Names (Steinberger et al., 2011). Here, we provide more details on the fundamental differences WikiKGE-10 and JRC-Names, including: i) WikiKGE-10 is completely manually-created; ii) WikiKGE-10 is mapped 1-to-1 to Wikidata; iii) WikiKGE-10 is not limited to persons and organizations; iv) JRC-Names considers names with spelling mistakes as valid names (as they may appear in real-life scenarios), whereas WikiKGE-10 considers them incorrect (as our objective is to obtain a multilingual knowledge graph that is as clean as possible); v) JRC-Names does not distinguish between entities that have the same name, since it is "very likely that different persons sharing the same first and last name have the same identifier because no disambiguation mechanism is in place."

## C  Methodology: Addendum

In this section, we provide more details on the methods we investigate in our paper, namely, MT, WS, LLMs, and M-NTA.

### C.1  Contextualizing entity names

As mentioned in Section 4.1, converting entity names from one language to another – by using machine translation, looking them up with Web search engines, or querying language models – is challenging because entity names can be ambiguous. Therefore, we contextualize entity names before converting them from one language to another language, i.e., we add information that a system can use to disambiguate an entity name and produce the correct output in the target language.

More specifically, given the fact that we already know the entity identifier associated to the entity

---

[5]The original paper on Krippendorff's alpha suggests that tentative conclusions can be made with a score greater than 0.67 and strong conclusions can be made with a score greater than 0.80.

name we would like to translate, we retrieve its corresponding description from Wikidata in the same language as the entity name, and use it to form a pseudo-natural language sentence.[6] For example, the entity name *Apple* is contextualized as "*Apple is an American technology company*" and "*Apple is a fruit of the apple tree*" depending on whether it corresponds to entity Q312 or Q89, respectively. In case of missing entity descriptions for a target language, we construct a simple entity description starting from its instance-of statements in Wikidata, e.g., "*Albert Einstein is a human.*" While more complex strategies or more relations may be used to better contextualize entity names, devising more complex strategies – which may require separate ad hoc solutions for MT, WS, and LLMs – is beyond the scope of this paper. We leave the investigation of more complex techniques for entity name contextualization to future work.

## C.2 Aligning and de-contextualizing entity names

While the advantage of contextualizing entity names is evident, the main disadvantage is that system will "translate" an entity name and also its contextualization information, possibly mixing the two types of textual information. This issue is particularly relevant when translating to a target language with a syntax that is significantly different from the source language or to a target language with non-trivial segmentation rules, e.g., from English to Japanese or Chinese. Therefore, we need to de-contextualize the translated name, i.e., we need to align the translated name to the original name and remove the contextualization information that was translated together with the name.

To address this issue, we follow recent studies (Chen et al., 2022) in alignment techniques, which show that MT is surprisingly robust to the insertion of symbols in the input sentence. More specifically, we indicate the start and the end of the entity name in the input sentence with special markers; for example, "*[Apple] is an American technology company.*" After translating the contextualized entity name into the target language, we detect the start and end markers in the translation and use their position to extract the translated entity name. Our analysis reveals that such an alignment system produces valid alignments most of the time in a

subset of manually-inspected instances. While this alignment system can be replaced by more complex alignment techniques, our analysis suggests that alignment errors are not the primary factor in end-to-end evaluation; we measured the number of errors attributable to misalignments and found that only 2% of the translated sentences contains such errors. Therefore, we can conclude that alignment errors are not a major bottleneck to end-to-end performance on WikiKGE-10 – probably due to the simplicity of the syntactic structure of the sentences that result from the contextualization process – and leave the investigation of more complex alignment systems to future work.

## C.3 MT: implementation details

In our experiments with MT, we decided to limit the number of source languages to 7, namely, German, English, Spanish, French, Italian, Japanese, and Chinese. The main reason behind this choice is that the quality translations from automatic systems has been shown to still lag behind when the source language is a lower-resource language, e.g., Korean. Therefore, in this work, we focus our attention on higher-resource languages for which MT has been proven to achieve satisfactory results on several standard benchmarks, allowing us to iterate faster. We hypothesize that translating from lower-resource languages does not result in performance that is significantly better than what we can see in Table 2, even when the linguistic families of the source and target languages are close. However, we leave an investigation on the effect of carefully choosing source-target language pairs for MT to future work.

## C.4 WS: implementation details

In Section 4.1, we discussed how WS can be used to retrieve entity names in a target language: given an entity name in a source language $l_s$, we can perform a search using a query like "*[entity-name] in [$l_t$]*" to obtain results in a target language $l_t$. Moreover, we can enrich the query by adding contextual information in the form of Wikidata descriptions, as discussed in section C.1, resulting in enriched queries like "*[entity-name] ([entity-description]) in [$l_t$]*" to mitigate the problem of ambiguous names, e.g., not only there are more than 10 people in Wikipedia that could be referred to as *Michael Jordan* but also songs and movies.

More specifically, given an entity $e$ and one of its names $e_s^n$ and its Wikidata description $e_s^d$ in a

---

[6]Wikidata descriptions can be retrieved from the Wikidata dump. Each entity may have multiple Wikidata descriptions, one for each language if available.

source language $l_s$, we build a search query as described above, limiting the choice of $l_s$ to English. Then, we parse the HTML response and collect the most frequently highlighted terms, i.e., those terms that are in bold (between  tags) or emphasized (between  tags), in the top-10 websites returned by the search engine. Finally, we keep the top-5 entity names retrieved from the collected terms if they appear at least 2 times among the highlighted results. As discussed in Section 5.2, such an approach – even though it tries to imitate how humans look up information on the Web – results in a significant amount of noise due to the collection of a significant number of terms that are only semantically-related to the query and not semantic matches.

## C.5    LLMs: implementation details

In our experiments, we investigate two main LLMs: mT0 and GPT. The former is the instruction-finetuned version of mT5, a state-of-the-art multilingual LLM. For mT5, we take into account three variants – large, xl, and xxl – which differ in their size to investigate if and to what extent increasing the number of trainable parameters in a language model is beneficial for the task under consideration.

For our experiments, we evaluate the effectiveness of mT5 and GPT with one-shot prompts, i.e., we provide a description of the task and one example of input/output to the LLM before requiring them to generate the entity name of interest. More specifically, each prompt is constructed as follows:

- Task definition: given an entity name in English and a short description of the entity in English, complete the following with the corresponding entity name in [$l_t$].

- Example:
    - English name: [$\hat{e}_s^n$]
    - English description: [$\hat{e}_s^d$]
    - [$l_t$] name: [$\hat{e}_t^n$]

- Task:
    - English name: [$e_s^n$]
    - English description: [$e_s^d$]
    - [$l_t$] name:

where $l_t$ is the target language, $\hat{e}$ is the entity used for the example, and $e$ is the entity of interest. We choose the example entity $\hat{e}$ at random from the top-10% entities with the only constraint that $\hat{e}$ and

$e$ have the same entity type, e.g., if we want to generate the name for $e$ and $e$ is a person, then also the example entity $\hat{e}$ shall be a person. Notwithstanding the input/output example provided, we observe that sometimes LLMs, even when they output correct names, do not conform to the same input/output format as the example, e.g., they add preambles ("the name of X is Y", "as a language model, I...") or explanations ("X because..."). This makes it hard to extract the relevant portion of text, resulting in alignment errors.

## C.6    M-NTA: implementation details

In this section, we provide more details on three important factors for the implementation of M-NTA, namely, the value of $\lambda$, the choice of $\phi$, and the individual contribution of each sub-system (MT, WS, and LLMs) in M-NTA.

### C.6.1    The value of $\lambda$

In Section 4.2, we introduced M-NTA, our novel approach to combine MT, WS, and LLMs, and described how it scores and ranks the answers $Y = \{y : \sigma(y) \geq \lambda\}$ according to a threshold hyperparameter $\lambda$, mentioning that $\lambda = 2$ is the most robust choice for coverage. Here, we expand our discussion on $\lambda$, showing how the choice of its value can significantly vary the precision and recall of the answers provided by M-NTA.

At a high level, the intuition behind $\lambda$ is that it is a hyperparameter that controls the number of "supporting evidences" required by M-NTA to consider an answer as plausible; on the contrary, if an answer is supported by fewer than $\lambda$ evidences, then M-NTA considers such an answer as noise. Therefore, we can expect that increasing the value of $\lambda$ will result in more precise predictions at the cost of recall, and decreasing the value of $\lambda$ will result in more broad coverage but also less precise answers. This is indeed the case in our experiments, as we can see in Figures 6 and 7, in which we can observe that increasing the value of $\lambda$ decreases the overall recall while increasing the precision of the answers on a sample of the Italian and Korean test sets of WikiKGE-10. Given the results of M-NTA for different values of $\lambda$ across the 10 languages of WikiKGE-10, we observed that $\lambda = 2$ is empirically the best choice on average if we want to balance precision and recall in coverage. However, we also note that the decision about the value of $\lambda$ can be also affected by the downstream application of interest: if the use case is adding textual

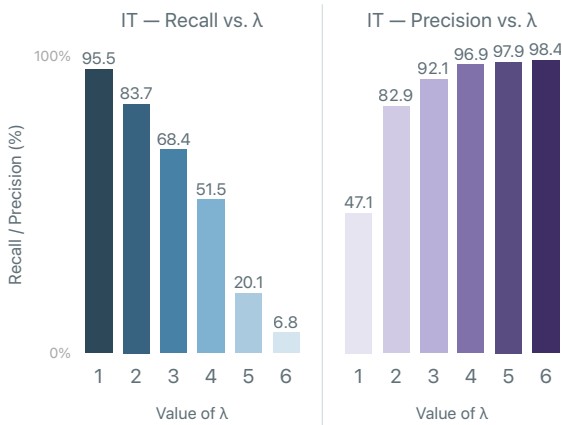

Figure 6: Recall and Precision (%) of M-NTA in the Italian test set of WikiKGE-10 (coverage) for increasing values of $\lambda$, ranging from 1 to 6. We can observe how the Recall decreases as the Precision increases.

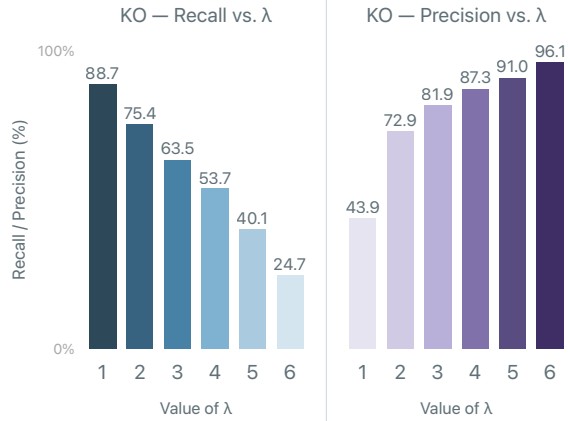

Figure 7: Recall and Precision (%) of M-NTA in the Korean test set of WikiKGE-10 (coverage) for increasing values of $\lambda$, ranging from 1 to 6. We can notice the same trend as in Figure 6.

information to a knowledge graph for direct user consumption, then we may want to prefer precision over recall and increase the value of $\lambda$ accordingly; otherwise, if we want to use textual information for the creation of multilingual embeddings, then we may be more interested in recall for covering as many entities as possible.

### C.6.2 The choice of $\phi$

One important factor in the design of M-NTA is the choice of the function $\phi(y, y') \rightarrow \{0, 1\}$, which establishes whether an answer $y$ from a system $f(\cdot)$ is supported by the answer $y'$ from another system $f'(\cdot)$. While $\phi$ can be any "similarity" metric, e.g., a measure of vector similarity, the final choice depends on the type of textual information represented by each answer. In this paper, we focus

|  | R | P | F1 |
|---|---|---|---|
| M-NTA$_{\lambda=1}$ | **89.7** | 42.4 | 57.1 |
| M-NTA$_{\lambda=2}$ | 71.4 | 81.5 | **75.6** |
| M-NTA$_{\lambda=3}$ | 56.4 | 82.3 | 66.5 |
| M-NTA$_{\lambda=4}$ | 41.9 | 87.3 | 56.0 |
| M-NTA$_{\lambda=5}$ | 23.2 | 90.3 | 35.5 |
| M-NTA$_{\lambda=6}$ | 10.4 | **93.8** | 18.0 |

Table 5: Recall (R), Precision (P), and F1 Score of M-NTA with different values of M-NTA across the 10 languages of WikiKGE-10. The value of $\lambda$ in M-NTA can be tuned to have broad recall or high precision.

|  | C | P |
|---|---|---|
| M-NTA$_{Full}$ | 53.6 | 79.9 |
| M-NTA$_{no-WS}$ | 53.2 | 79.8 |
| M-NTA$_{no-LLM}$ | 43.8 | 71.1 |
| M-NTA$_{no-WS/no-LLM}$ | 43.2 | 70.9 |

Table 6: Ablation study on the individual components of M-NTA on coverage (C) and precision (P). All results reported for M-NTA with $\lambda = 2$ and $\lambda = 1$ for precision.

on entity names, for which even a slight variation between two names can mark the difference between a correct name and an incorrect one, e.g., *Olivier* and *Oliver*. Therefore, we choose exact match between lower-cased, punctuation-stripped entity names as the function $\phi$, i.e., a name $y$ is supported by another name $y'$ if and only if $y = y'$, except for letter casing (e.g., *Canary* and *canary*) and punctuation (*Michael B Jordan* and *Michael B. Jordan*). As we will see in section C.7, other forms of $\phi$ may be more appropriate for types of textual information different from entity names.

### C.6.3 Ablation study

Throughout the paper, we mentioned multiple times that the main strength of M-NTA is its capability to combine the answers provided by MT, WS, and LLMs. Here, we carry out an ablation study to quantify and better understand the individual impact of each subsystem in M-NTA. More specifically, we compare the results of the "full" M-NTA to M-NTA without Google Web Search (M-NTA$_{no-WS}$), without GPT-3.5 (M-NTA$_{no-LLM}$), and only with MT from 7 languages (M-NTA$_{no-WS/no-LLM}$). As we can see in Table 6, even when M-NTA does not rely on answers from WS and LLMs, the results of M-NTA$_{no-WS/no-LLM}$

| Entity | Source | Entity description |
|---|---|---|
| *Bufuri* 
 Q1002164 | Wikidata 
 M-NTA | "company" 
 "car manufacturer" |
| *Reinhard Zöllner* 
 Q100502 | Wikidata 
 M-NTA | "German historian" 
 "university professor and German historian" |
| *Haukadalur* 
 Q1034430 | Wikidata 
 M-NTA | "valley" 
 "valley in Iceland with a geothermal area" |
| *Paola Cortellesi* 
 Q1042721 | Wikidata 
 M-NTA | "Italian actress and singer" 
 "Italian actress, screenwriter, television author, comedian and singer (1973-)" |
| *Washuzan Highland* 
 Q10345405 | Wikidata 
 M-NTA | "Japanese amusement park in Okayama prefecture" 
 "An amusement park in Shimotsui, Kurashiki City, Okayama Prefecture" |

Table 7: Selected examples of descriptions generated by M-NTA compared to Wikidata. As you can see, M-NTA is able to improve descriptions in high-resource languages. As we can see in this Table, M-NTA is able to provide important information in the description.

### C.7 Applying M-NTA to entity descriptions

While the focus of WikiKGE-10 is on entity names, the approaches described in Section 4.1 – MT, WS, and LLMs – and M-NTA can also be applied to other types of textual information. As discussed in sections 2 and 3, entity descriptions are another popular type of textual information used in recent approaches. In this section, we describe how MT and M-NTA can be easily adapted to convert entity descriptions from one language to another, while we leave a more in-depth study about the effectiveness of WS and LLMs for entity descriptions to future work.

Adapting the MT-based approach to generate entity descriptions in a target language is straightforward. In section C.2, we discussed the necessity of using special markers to facilitate the extraction of the translated entity name from the translated sentence, e.g., "*[Apple] is an American multinational technology company*." To extract the entity description instead of the entity name, we can simply place the special markers around the entity description, e.g., "*Apple is an [American multinational technology company]*." Thanks to this simple modification, the rest of the pipeline for the MT-based approach can remain the same.

Adapting M-NTA to generate entity descriptions in a target language requires an additional step, i.e., designing an appropriate function $\phi(y, y') \to \{0, 1\}$ (see section 4.2) to establish when a description $y' = \bar{e}_t^d$ counts as supporting evidence for a different description $y = e_t^d$. Indeed, a description may imply another description even if they are not exact matches. For example, the English description for *Earth* (Q2) is "third planet from the Sun in the Solar System", which implies the Spanish description "planet in the Solar System, third by distance from the Sun" (translated in English from Spanish). To address this issue, we define $\phi$ as follows:

$$\phi(y, y') = \begin{cases} 1 & \text{if } \text{sim}(y, y') > 0.5 \\ 0 & \text{if } \text{sim}(y, y') \le 0.5 \end{cases}$$

where $\text{sim}(\cdot)$ is the cosine similarity between the vector representations of $y$ and $y'$. We compute the vector representations of the descriptions by using XLM-RoBERTa (base) (Conneau et al., 2020).

## D WikiKGE-10: Additional Results

In this section, we provide additional results to complement the main results described in Section 5.

Indeed, it is interesting to observe how the results would change if we slightly relax the metrics of coverage and precision. In particular, we relax coverage to provide a positive score in case a system is able to provide at least one valid entity name for a given entity. Similarly, we relax precision to provide a positive score in case a system is able to identify at least one invalid entity name for a given entity. Table 8 provides an overview of the results. As one could expect, the scores on coverage increase, as it is easier to provide one valid

| | | AR | | DE | | EN | | ES | | FR | | IT | | JA | | KO | | RU | | ZH | | Avg | |
|---|---|---|---|---|---|---|---|---|---|---|---|---|---|---|---|---|---|---|---|---|---|---|---|
| | | C | P | C | P | C | P | C | P | C | P | C | P | C | P | C | P | C | P | C | P | C | P |
| *MT from* | DE → | 57.8 | 35.4 | – | – | 62.7 | 36.9 | 68.3 | 36.4 | 66.5 | 40.1 | 72.7 | 47.5 | 44.2 | 25.8 | 60.7 | 38.9 | 59.5 | 35.0 | 46.9 | 26.7 | 59.9 | 35.9 |
| | EN → | 71.4 | 49.9 | 76.6 | 60.7 | – | – | 81.9 | 53.3 | 74.8 | 52.6 | 78.9 | 59.5 | 51.5 | 38.6 | 70.8 | 54.1 | 67.8 | 47.6 | 55.5 | 42.7 | 69.9 | 51.0 |
| | ES → | 57.5 | 32.9 | 59.8 | 38.9 | 57.7 | 33.2 | – | – | 66.3 | 39.9 | 68.6 | 47.0 | 43.7 | 24.1 | 61.1 | 37.2 | 56.2 | 30.7 | 46.1 | 26.4 | 57.5 | 34.5 |
| | FR → | 61.1 | 33.9 | 67.0 | 41.2 | 63.1 | 33.7 | 72.1 | 35.4 | – | – | 71.9 | 46.0 | 47.8 | 26.6 | 62.8 | 37.0 | 59.2 | 29.9 | 47.3 | 25.7 | 61.4 | 34.4 |
| | IT → | 58.9 | 28.4 | 64.9 | 37.5 | 59.1 | 30.5 | 70.2 | 31.8 | 66.9 | 34.3 | – | – | 43.3 | 20.8 | 59.9 | 30.8 | 58.7 | 26.6 | 46.4 | 21.8 | 58.7 | 29.2 |
| | JA → | 40.9 | 16.7 | 32.0 | 12.4 | 25.7 | 9.0 | 37.8 | 11.6 | 33.8 | 10.7 | 38.0 | 14.0 | – | – | 56.4 | 33.8 | 34.9 | 11.7 | 43.4 | 24.6 | 38.1 | 16.1 |
| | ZH → | 38.0 | 16.5 | 27.6 | 9.6 | 20.3 | 6.4 | 30.2 | 9.0 | 30.2 | 9.7 | 30.2 | 10.5 | 33.1 | 17.6 | 47.7 | 27.3 | 30.6 | 9.8 | – | – | 32.0 | 12.9 |
| *WS* | Google_Search | 52.2 | 41.4 | 58.8 | 55.1 | – | – | 69.9 | 47.2 | 58.2 | 46.4 | 66.1 | 58.2 | 34.2 | 22.2 | 45.3 | 37.7 | 42.1 | 42.7 | 39.4 | 31.8 | 51.8 | 42.5 |
| *LLMs* | mT0_1B | 49.1 | 37.2 | 56.2 | 44.3 | – | – | 70.7 | 45.1 | 59.1 | 44.1 | 65.1 | 57.2 | 36.3 | 18.2 | 45.0 | 38.9 | 44.1 | 38.0 | 37.2 | 31.0 | 51.4 | 39.3 |
| | mT0_3B | 53.2 | 38.1 | 57.2 | 46.6 | – | – | 71.8 | 45.2 | 61.0 | 44.6 | 65.9 | 58.1 | 38.1 | 19.2 | 46.4 | 38.8 | 46.0 | 38.6 | 39.5 | 32.0 | 53.6 | 40.1 |
| | mT0_7B | 54.2 | 40.2 | 59.1 | 50.1 | – | – | 74.4 | 47.8 | 62.2 | 47.2 | 69.4 | 57.9 | 39.2 | 23.4 | 48.0 | 40.1 | 46.1 | 39.1 | 41.2 | 32.5 | 54.7 | 42.1 |
| | GPT-3.5 | 67.1 | 48.8 | 75.9 | 61.3 | – | – | 80.3 | 57.6 | 77.1 | 54.2 | 76.4 | 57.3 | 54.4 | 41.0 | 73.3 | 52.2 | 69.1 | 44.4 | 57.2 | 44.1 | 70.7 | 51.2 |
| | M-NTA $_{GPT\text{-}3.5}$ | **75.9** | **70.6** | **79.7** | **73.1** | **67.8** | **58.1** | **86.2** | **69.1** | **83.2** | **66.3** | **87.1** | **74.2** | **59.6** | **51.9** | **79.1** | **75.5** | **75.2** | **64.2** | **62.7** | **60.5** | **75.6** | **66.3** |

Table 8: F1 scores on entity names coverage (C) and precision (P) in WikiKGE-10 when identifying at least one valid name for coverage and at least one invalid name for precision. The symbol "–" is used to indicate that source and target languages are the same. Best results in **bold**.

| | Entities | | | Names | | |
|---|---|---|---|---|---|---|
| | coverage (%) | | | coverage (%) | | |
| | W | +M-NTA | Δ | W | +M-NTA | Δ |
| DE | 94.63 | 97.42 | + 2.79 | 95.12 | 96.45 | +1.33 |
| EN | 99.09 | 99.23 | + 0.14 | 93.48 | 93.88 | +0.40 |
| ES | 95.01 | 97.10 | + 2.09 | 93.12 | 94.39 | +1.27 |
| FR | 96.07 | 97.64 | + 1.57 | 96.13 | 97.03 | +0.90 |
| IT | 93.07 | 96.80 | + 3.73 | 95.52 | 97.75 | +2.23 |
| JA | 87.52 | 91.53 | + 4.01 | 91.88 | 94.15 | +2.27 |
| ZH | 54.15 | 64.44 | +10.29 | 55.60 | 64.91 | +9.31 |
| Avg | 88.51 | **92.02** | + 3.52 | 88.69 | **91.22** | +2.53 |

Table 9: Comparison of Wikidata (W) and Wikidata + M-NTA (+M-NTA) on entity and name coverage for entity-type queries in MKQA.

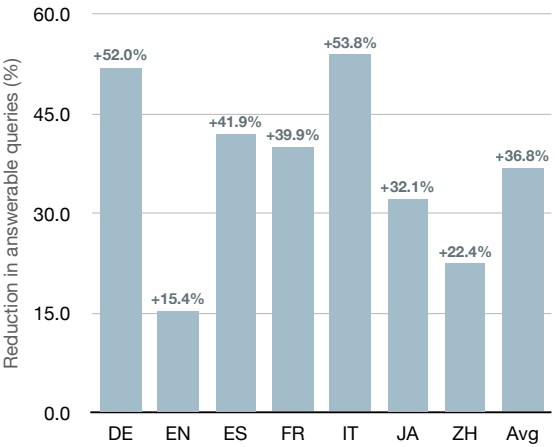

Figure 8: Reduction rate in the number of unanswerable queries in MKQA when using M-NTA to improve the coverage of Wikidata.

name for an entity instead of the complete list of valid names. However, the performance in precision decrease usually decreases, as we hypothesize that there are entities for which it is more difficult to identify incorrect entity names.

# E Impact on Downstream Tasks: Question Answering

In section 6, we have investigated the impact of increasing coverage and precision of textual information in two downstream tasks, namely, multilingual entity linking and multilingual knowledge graph completion. Here, we also investigate the impact of our work on Question Answering (QA), with a specific focus on knowledge-seeking queries. One of the main characteristics of knowledge-seeking queries is that they can be usually answered by navigating a knowledge graph and returning (the name of) an entity, e.g., the answer to the query "What is the highest mountain in Washington, US?" is *Mount Rainier* (Q194057). However, if the knowledge graph does not provide a lexicalization for the entity in the target language, then a knowledge-based QA system will not be able to provide a correct answer. Therefore, increasing the coverage of entity names across languages is essential to extend the support of knowledge-based QA systems to multilingual settings.

To quantify the impact of M-NTA on QA, we consider the subset of queries in MKQA (Longpre et al., 2021), a multilingual QA dataset for knowledge-seeking queries, whose type of answer is classified as "entity", i.e., those queries that can be answered by providing the name of a Wikidata entity. Importantly, the original authors of MKQA manually added names (primary names and aliases) for all those Wikidata entities that did not have a lexicalization. Therefore, there is a set of questions in MKQA which are "unanswerable" by a knowledge-based QA system that relies on Wikidata; this set of unanswer-

able questions impose an upper bound to the results achievable by any knowledge-based QA system. More specifically, we measure the number of answerable/unanswerable queries when relying only on Wikidata[7] compared to using an M-NTA-augmented Wikidata (Wikidata + M-NTA) in two settings:

- Entity coverage: the number of entities in the answers of MKQA for which Wikidata (or Wikidata + M-NTA) can provide at least one name;

- Name coverage: the number of names for the entities in the answers of MKQA that are also present in Wikidata (or Wikidata + M-NTA).

As we can see in Table 9, using M-NTA allows us to increase the number of answerable queries both when we look at entity coverage (+3.52% absolute improvement) and name coverage (+2.53% absolute improvement). Notably, M-NTA provides a significant increase in entity coverage for simplified Chinese (+10.29% absolute improvement), which is the language with lowest coverage, but also in English (+0.14% absolute improvement). Although the absolute improvement in English seems small, entity coverage in English is already high in Wikidata (99.09%): another way to look at this improvement is by analyzing the reduction rate in the number of unanswerable queries. As we can see in Figure 8, the reduction rate in the number of unanswerable queries in MKQA can be reduced significantly when using M-NTA to improve the coverage of Wikidata. Even for English, the reduction rate is about 15.4%, which becomes as high as 52.0% and 53.8% in German and Italian, respectively.

---

[7]As of April 2023.