# OpenReview forum: "Increasing Coverage and Precision of Textual Information in Multilingual Knowledge Graphs"
_EMNLP/2023/Conference — EMNLP 2023 Main_

### Official Review · Reviewer_N39A · 2023-08-02

**Soundness:** 3

**Excitement:**

3: Ambivalent: It has merits (e.g., it reports state-of-the-art results, the idea is nice), but there are key weaknesses (e.g., it describes incremental work), and it can significantly benefit from another round of revision. However, I won't object to accepting it if my co-reviewers champion it.

**Missing References:**

- Generating Multilingual Descriptions from Linguistically Annotated OWL Ontologies: the NaturalOWL System (Galanis and Androutsopoulos, 2007)
- Exploiting OWL Ontologies in the Multilingual Generation of Object Descriptions (Androutsopoulos et al 2005)
- JRC-NAMES: A Freely Available, Highly Multilingual Named Entity Resource (Steinberger et al, 2011)

**Paper Topic And Main Contributions:**

This work is on the task of obtaining multilingual names/labels for entities present in a knowledge graph, which can be useful for plugging such missing information in large knowledge graphs (that are typically only well-populated for languages where Wikipedia articles and editors number the most, e.g. English). The authors collected a human-verified dataset of approximately 35,000 multilingual names for 10 languages across different typologies and used this for their evaluation. The authors propose a method for the task by combining three means to obtain these multilingual names (machine translation, web search and prompting large language models), and which ensures that only name/label candidates that is supported by another method is accepted. They report increase in coverage and precision (see questions below) from their method, and also carried out experiments with multilingual entity linking and multilingual knowledge graph completion to show that adding predictions from their method improves these downstream tasks.

**Questions For The Authors:**

- A - What platform did you use for collecting the annotations? How did you compute the pairwise inter-annotator agreement values you report in Figure 2 of Appendix A2? Is it the average of all pairwise comparisons? How many annotators were involved in the annotation process and what is the average number of annotations they did? Line 333 states that besides rating the multilingual entity name/labels, the human graders also added new valid names. How were these added names validated? Were they put into the annotation pool for other annotators to verify too? Does the line "correct not in Wikidata" in Table 1 refer to these added names?
- B - How exactly do you compute the coverage and precision in Table 2? i.e. The increase in coverage by M-NTA for each language is measured with respect to what? The paper will also benefit from at least a few lines clearly stating how this evaluation was done (e.g. how the train test splits were done, random split of names or random split of entities etc).
- C - Section 4.1. Your method seems to rely on Wikidata text descriptions (Line 440), which you also state are significantly missing in Wikidata, can other types of text be used instead?

**Reasons To Accept:**

The human-verified evaluation dataset collected by the authors can be useful to supplement another multilingual entity name/label dataset that was released in 2011 (Steinberger et al, 2017) by introducing new and more recent entities for evaluation.  An automatic method to help plug the many missing multilingual entity names in knowledge bases can be helpful, especially for downstream knowledge base QA models to use. More immediately, it can also allow wider access to/understanding of such knowledge graph information by non-English speakers.

**Reasons To Reject:**

The claims made in parts of the the abstract, introduction and conclusion do not fully match with the experimental work carried out (see points 1,2,4 in **Typos Grammar Style And Presentation Improvements** below). There are questions about the procedures for obtaining the paper's experimental results. For example, it is not clear how the coverage and precision scores in Table 2 were computed.

**Reproducibility:**

4: Could mostly reproduce the results, but there may be some variation because of sample variance or minor variations in their interpretation of the protocol or method.

**Reviewer Confidence:**

3: Pretty sure, but there's a chance I missed something. Although I have a good feel for this area in general, I did not carefully check the paper's details, e.g., the math, experimental design, or novelty.

**Typos Grammar Style And Presentation Improvements:**

- The task of generating names and entity descriptions have meaningful differences. Compared to multilingual entity name extraction/retrieval/generation (which is the subject of the new benchmark dataset collected and the experimental work in this paper), the latter involves additional NLG subtasks such as content selection (what information to realize in the text) and surface realisation (for grammaticality and fluency). The main body of the paper currently refers heavily to the entity description task, but does not contain any data or experimental tasks that investigates this task. Some suggestions: It will be better for the reader if you clearly refer to the task you investigated (e.g. multilingual entity name translation/retrieval/generation) instead of using the term "textual description" throughout, which can be read as including the "entity description" task. The "entity description" task could then be introduced and discussed on a future work/directions section nearer the conclusion. The paper could also benefit from a title that more clearly describes the actual task you investigate (it is too broad and general currently).
- Line 365: Better to point the reader to the precise subsection (i.e. appendix B7). Also, it does not seem accurate to say "but we also **show** that the methodologies discussed in this section can be extended to other types of textual information, such as entity descriptions...", since the experiments and results in the following sections are only on multilingual entity name retrieval/extraction/generation task.  Perhaps you mean "we also suggest some ways the methodologies..."
- Line 549: This paragraph can benefit from more clarity as it appears at odds with other parts of the paper. In Section 3.4 you state that WikiKGE-10 was created by querying top-10% head entities, but in this section you refer to long-tail entities. It is also not clear what you mean by "the risk of using them is higher in case of error". What is the specific risk, and are you able to point to the parts of your experimental results that supports this claim?
- Line 027, 104 and 547: Although you include (in the appendix) a study about how using M-NTA can reduce the number of questions in the MKQA dataset that require human manual annotation to obtain multilingual entity names as answers, maybe this may be better characterised as a method for data improvement rather than the actual task of question answering?
- Line 475: it may be better to state and describe your choice of phi (the similarity measure) here, or at least point the reader to the appendix where you mention that it is exact string match (after lowercasing and stripping punctuation).

---

> ### Author Rebuttal · Authors · 2023-08-29
>
> We really appreciate the time that went into writing this review; we hope this answer can clarify our work and address your concerns about the soundness of our work!
>
> __Questions For The Authors__
>
> * __A - questions about the annotation process__
>     * __“What platform did you use for collecting the annotations?”__ We used an internal annotation tool with an interface that we designed specifically for this task. In case of acceptance, we will expand Appendix A with screenshots that show the same UI that the annotators used. Using our own annotation platform allows us to have better control over the annotators, including training them on the task using a separate dataset, checking their language qualifications, possibility to directly reach out to an annotator, custom entrance tests, regular quality checks, etc.
>     * __“How did you compute the pairwise inter-annotator agreement values you report in Figure 2 of Appendix A2?”__ We report pairwise inter-annotator agreement using (an extension of) Cohen’s Kappa: each item was annotated by 3 annotators, and we computed the agreement by considering all rating pairs. Although Cohen’s Kappa is usually used to compute IAA between 2 raters, its extension to $n > 2$ raters, known as Fleiss’ Kappa, can sometimes lead to paradoxical results (*“Handbook of Interrater Reliability”*, Kilem Gwet) and it’s not widely adopted by the NLP community. Instead, we also report IAA using Krippendorff’s Alpha (Figure 3), which can be applied to $n > 2$ raters and also takes into account the cardinality of the ratings.
>     * __“How many annotators were involved in the annotation process and what is the average number of annotations they did?”__ The number of annotators varied from language to language. On average, we employed 21 annotators per language; each rater annotated around 420 items (+50 for training).
>     * __“How were these added names validated? Were they put into the annotation pool for other annotators to verify too?”__ Yes, new names were re-validated by 3 annotators using the same process.
>
> * __B - questions about the evaluation__
>     * __“How exactly do you compute the coverage and precision in Table 2?”__ We will expand our description of coverage and precision with a formal definition in the extra page of the camera ready in case of acceptance. In our case, we say that a method “covers” an entity $e$ in a language $l$, if it is able to provide a valid entity name for $e$ in $l$ (exact string matching with lower casing and punctuation stripping). The coverage score indicates the percentage of entities covered by a given method in $l$. Similarly, the precision score indicates the percentage of cases in which a given method can identify that an entity name in Wikidata is actually incorrect (i.e., a name that is considered incorrect by our annotators in WikiKGE-10).
>     * __“how the train test splits were done, random split of names or random split of entities?”__ We note that all the methods we investigate (even our method M-NTA) are unsupervised, that is, they are not trained on a task-specific training set. For what concerns our test set, we mention the sampling process in Section 3.4: we created our test set (WikiKGE-10) by randomly sampling 1000 entities from the top-10% of the popular entities in Wikidata, sorted by the number of page views in each language of interest.
>
> * __C - questions about the methodology__
>     * __“[naturalization in M-NTA] can other types of text be used instead?”__ Yes! As M-NTA already yields positive results, we left the investigation of more complex methods to future work. However, the naturalization step of M-NTA can be immediately improved: for example, one could use graph-to-text approaches for cases in which an entity description is not available.
>
>
> __Typos, Grammar Style And Presentation Improvements__
>
> We would also like to address the following points (“the claims made in parts of the the abstract, introduction and conclusion do not fully match with the experimental work carried out”), hoping to clarify the soundness of our work.
>
> * __“the term "textual description" throughout [the paper], which can be read as including the "entity description" task"__ We use the term "textual description" only once in the paper and with the same meaning as "entity description". If the Reviewer refers to "textual information", we use this term to refer to language-dependent "string" information in a KG. Although our paper does not include an intrinsic evaluation of entity descriptions, we still describe how our method can be applied to them (Appendix B.7), provide a qualitative analysis with some output examples (Table 8), and — most importantly — include an extrinsic evaluation of their downstream impact (see next answer).
> * __“[the paper] does not contain any data or experimental tasks that investigates this task [entity descriptions]”__ Our paper does contain experiments that involve entity descriptions. In our experiments on MKGC (Section 5), the model also uses entity descriptions to create entity representations (line 610). It is interesting to notice that the dataset used for MKGC is constructed to have complete coverage of entity names; this suggests the improvements come from better coverage (and quality) of the entity descriptions rather than the names. We will expand the camera ready with this information and examples, hoping to make it clearer that our experiments extrinsically evaluate the impact of better entity descriptions.
> * __“the experiments and results in the following sections are only on multilingual entity name[s]”__ Please, see answer above.
> * __“maybe this may be better characterised as a method for data improvement rather than the actual task of question answering?”__ Indeed, our contributions fall into the category of data-centric AI approaches (lines 77—81; 272-280; 315-317; 627-633). In fact, our work is a way to improve the data (KG), and we show that our proposed methodology for improving textual data in a KG is a simple and effective way for better downstream results in three tasks without modifying the models.
> * __“What is the specific risk [of using LLMs], and are you able to point to the parts of your experimental results that supports this claim?”__ When comparing the mistakes of two systems, e.g., LLMs and web search, a metric may hide the relative gravity of the two mistakes, and we believe that it is important for our readers to be aware of this. More specifically, our qualitative analysis suggests that a mistake made by web search (which usually returns semantically similar results) may be less problematic than a mistake made by an LLM (which can generate completely unrelated and erroneous results more easily). Therefore, depending on the downstream application of interest, we argue that one may still prefer web search to BLOOMZ even if the former slightly underperforms the latter in terms of numerical results in our benchmark to avoid the cases shown in Section 4. Finally, we note that we did not limit our analysis to the entities in the top-10%, hence, we also mention long-tail entities.
>
> __References__
>
> Thank you for pointing out those papers! We will extend the Related Work to include them and describe the high-level differences with our work, especially between JRC-Names and WikiKGE-10.
>
> In short: i) WikiKGE-10 is completely manual; ii) WikiKGE-10 is mapped 1-to-1 to Wikidata; iii) WikiKGE-10 is not limited to persons and organizations; iv) JRC-Names considers names with spelling mistakes as valid names (as they may appear in real-life scenarios), whereas WikiKGE-10 considers them incorrect (as we aim for a clean KG); v) JRC-Names does not distinguish between entities that have the same name (“very likely that different persons sharing the same first and last name have the same identifier because no disambiguation mechanism is in place”).

---

### Official Review · Reviewer_JXeC · 2023-08-04

**Typos Grammar Style And Presentation Improvements:** Line 078 - The acronym AI should be i…
**Soundness:** 4

**Excitement:**

4: Strong: This paper deepens the understanding of some phenomenon or lowers the barriers to an existing research direction.

**Paper Topic And Main Contributions:**

Authors propose a methodology for knowledge graph enhancement (KGE), intended as the task of increasing coverage and precision of textual information, specifically for non-English languages and information.
WikiKGE-10, a resource for benchmarking data-centric-AI approaches on KGE of entity names in 10 languages, is presented.
Starting from the limitations of three well-known approaches dealing with KGE (machine translation, Web search, and large language models), authors propose M-NTA: Multi-source Naturalization, Translation, and Alignment, an approach which exploits all the three aforementioned methods.

It would be better specifying the version of ChatGPT used in the experiment (even better specifying the period in which the tests have been conducted).


**Reasons To Accept:**

The paper is well-written and motivated and represents a contribution to the field.
The aim of covering non-English languages is always a plus.


**Reasons To Reject:**

I do not see any reason for not accepting the paper.

**Reproducibility:**

4: Could mostly reproduce the results, but there may be some variation because of sample variance or minor variations in their interpretation of the protocol or method.

**Reviewer Confidence:**

4: Quite sure. I tried to check the important points carefully. It's unlikely, though conceivable, that I missed something that should affect my ratings.

---

> ### Author Rebuttal · Authors · 2023-08-29
>
> Thank you for your positive review: we really appreciate that you also believe *“covering non-English languages is always a plus”*!
>
> We will make sure to include the version of ChatGPT (which we used between March and May) in the camera ready of the paper.

---

### Official Review · Reviewer_sJ2r · 2023-08-05

**Soundness:** 5

**Excitement:**

4: Strong: This paper deepens the understanding of some phenomenon or lowers the barriers to an existing research direction.

**Paper Topic And Main Contributions:**

Problem: Knowledge graphs exhibit significant disparities in quality and data sparsity when dealing with non-English languages.
- The authors first explore this phenomenon for Wikidata and report lower coverage numbers for 9 different languages when compared to English
- The authors then create an annotated dataset called WikiKGE-10 that encompass 1000 entities listing their mentions (missing and otherwise) along with errors
- To improve multilingual coverage & accuracy for knowledge graphs (they call this Knowledge Graph Enhancement/KGE), the authors propose a method they term M-NTA as an ensemble of methods from machine translation, web search & language models
- They evaluate each of the above methods independently and compare them with that of their proposed M-NTA method for the WikiKGE-10 dataset
- Finally, they evaluate the value of an enhanced Knowledge Graph for Knowledge Graph Completion & Question Answering

Along the way, the authors clearly express their motivations backed by exploratory analysis and follow it up with a proposed solution backed by thorough experimental evidence.

**Questions For The Authors:**

Just double checking the interpretations I've made

- 324-330 - this specifies that the 1000 entities are likely different (barring some overlap) across languages. Is this the case? Or are they the same 1000 entities across all languages?

- Table 2: For a given column say AR, is it fair to think the F1 score here for coverage implies that for each of the 1000 entities in the WikiKGE-10 for AR, their corresponding entities in English (or other languages in case of MT) are used as a starting point to translate them back to AR and then compare it with the ground truth?

(some clarification here can help make this more concrete)

**Reasons To Accept:**

- The authors recognize and empirically show disparity in knowledge graphs across languages. The exploratory analysis done here is solid featuring a diverse, reasonably sized sample set of languages.
- The authors creation of the WikiKGE-10 dataset is a valuable resource that is reasonably sized covering 10 diverse languages. The annotation process is described in good detail making it a strong, objective benchmark to evaluate systems against.
- Their proposed method of combining multiple processes alongside experimental valuation is thorough and indicates strong improvements. The accompanying analyses and insights are also valuable on their own to understand broader behavior of machine translation, web search & llms in the context of cross-lingual knowledge transfer
- Their downstream evaluation is also thorough corroborating their broader claim that enhanced multilingual knowledge graphs are useful
- Clearly highlighted limitations of their dataset and methodology
- Finally, a very comprehensive, detailed appendix that complements the main paper

Overall, this is a very well written paper that is very clear and focused in claims made and is corroborated with rigorous experimental results.

**Reasons To Reject:**

Nothing major that I can think of.

**Reproducibility:**

4: Could mostly reproduce the results, but there may be some variation because of sample variance or minor variations in their interpretation of the protocol or method.

**Reviewer Confidence:**

4: Quite sure. I tried to check the important points carefully. It's unlikely, though conceivable, that I missed something that should affect my ratings.

**Typos Grammar Style And Presentation Improvements:**

- 008 - typo - comparatively scarce*
- 211 - typo - their content*

---

> ### Author Rebuttal · Authors · 2023-08-29
>
> Thank you for your review! We are truly grateful that our contributions — our analysis, our manually-created multilingual benchmark, our proposed methodology, and its usefulness in downstream applications — have been recognized to such an extent. We hope that this response can further solidify your understanding of our work.
>
> __Questions__
>
> Your interpretations are correct! We will use the extra page in the camera ready to provide additional details. In the meanwhile, we hope you can find the following answers helpful.
>
> * __“the 1000 entities are likely different (barring some overlap) across languages”:__ Yes, the 1000 entities in WikiKGE-10 (our manually-curated benchmark) are different across languages.
>     * __Additional details.__ The difference between languages comes from two main factors: First, each sample of 1000 entities is drawn independently for each language. Second -- and most interesting in our opinion -- the entities in the top-10% are different in each language *l*, as we determine entity popularity according to the page views of an entity in *l*. Therefore, the composition of the entities in our benchmark is also affected by culture-specific aspects.
>
> * __“their corresponding entities in English (or other languages in case of MT) are used as a starting point”:__ Yes, the starting point is the existing Wikidata information in a source language. This information is “translated” using MT, LLM prompting, WS, or M-NTA into a target language. If the information of an entity *e* is correctly translated to the target language, then we say that *e* has been “covered”.
>     * __Additional details.__ Notice that a source language (even English) may have no information about some entities. In this case, MT/WS/LLMs will be unable to “translate” the information in any target language. This is exactly one of the strengths of our proposed approach (M-NTA), which can draw information from multiple languages instead of a single one. One could argue that LLMs can be prompted with information coming from multiple source languages too; we leave this investigation to future work, as it goes beyond a “resources and evaluation” paper, while noting that M-NTA can potentially improve further if we improve the underlying LLM.

---

### Meta-Review · Area_Chair_H6R7 · 2023-09-13

**Recommendation:** 5

**Metareview:**

This paper is concerned with knowledge graph enhancement to improve the coverage and precision; it specifically aims to improve entity information in 10 languages, and presents a large new dataset (35,000 multilingual names) as the WikiKGE-10 benchmark. The core idea of the approach is to combine previous approaches to this problem (leveraging machine translation, web search and LLMs) into their approach M-NTA "Multi-source Naturalization, Translation and Alignment".
The reviewers agree that the paper is well-written and sound; they particularly appreciate the crosslingual work and the evaluation, as well as the paper's clarity regarding limitations. Some more detailed points questions about the methods could be addressed in the rebuttal and should help to improve the final version of the paper.

---

### Decision · Program_Chairs · 2023-10-07

**Decision:**

Accept-Main

**Comment:**

This paper is concerned with knowledge graph enhancement to improve the coverage and precision; it specifically aims to improve entity information in 10 languages, and presents a large new dataset (35,000 multilingual names) as the WikiKGE-10 benchmark. The core idea of the approach is to combine previous approaches to this problem (leveraging machine translation, web search and LLMs) into their approach M-NTA "Multi-source Naturalization, Translation and Alignment".
The reviewers agree that the paper is well-written and sound; they particularly appreciate the crosslingual work and the evaluation, as well as the paper's clarity regarding limitations. Some more detailed points questions about the methods could be addressed in the rebuttal and should help to improve the final version of the paper.